# MEGA: A LARGE-SCALE MOLECULAR EDITING DATASET FOR GUIDED-ACTION OPTIMIZATION

## ABSTRACT

Large language models show strong potential for molecular editing, but progress has been constrained by the limited scale and quality of available training data. To address this, we introduce MEGA, a family of large-scale datasets comprising 72M molecule pairs, each representing a single property-improving chemical edit annotated with an explicit action: *Replace*, *Insert*, or *Delete*. We demonstrate MEGA's utility in a controlled supervised fine-tuning (SFT) setting, where a model trained on MEGA outperforms models trained on existing datasets by up to +21.47 percentage points in hit ratio. Furthermore, we show that Group Relative Policy Optimization (GRPO) post-training with a similarity-aware reward achieves state-of-the-art performance and a remarkable $\sim 36\times$ improvement in data efficiency, while also preserving edit locality. We release MEGA in open access to the community to enable data-centric benchmarks and accelerate progress in molecular editing with generative models.

## 1 INTRODUCTION

Molecular optimization is critical to drug discovery, guiding chemists in turning initial molecular hits into drug-like candidates. Unlike unconstrained molecule generation Gómez-Bombarelli et al. (2018); Jin et al. (2019), molecular editing involves targeted modifications, such as scaffold decoration, fragment substitutions, or precise structural refinements, that carefully balance therapeutic properties with chemical feasibility and synthetic practicality Seo et al. (2023); Jinsong et al. (2024).

To assist chemists in this iterative lead optimization process, recent approaches leverage large language models (LLMs), either through fine-tuning or by using them as reasoning agents capable of interpreting textual prompts (e.g. "increase solubility") and proposing relevant molecular edits Zimmermann et al. (2025); Mirza et al. (2025a). Additionally, reinforcement learning (RL)-based post-training can align these models even more closely with practical constraints, improving both chemical plausibility and edit precision Dai et al. (2025); Liu et al. (2025). Progress, however, is limited by data. Training and evaluating editing models requires goal-aligned edit datasets that pair a parent molecule with a proposed child and standardized outcomes, at a scale that supports both supervised fine-tuning and post-training Polykovskiy et al. (2020); Huang et al. (2021). Nevertheless, existing corpora either lack the scale required for robust training or omit explicit edit annotations needed for guided policy learning.

To close this gap, we curate MEGA (Molecular Editing with Guided Action), a large-scale, molecule editing dataset of (parent, child) molecule pairs spanning 28 tasks. The dataset is offered in in two scales, the primary MEGA dataset, containing 522 thousand successful edits, and an expanded version, MEGA-Large, with 31.4 million positive samples. We also release an additional 41 million valid and chemically close negative examples to enable contrastive learning and Reinforcement Learning (RL) reward shaping Robinson et al. (2021); Shen et al. (2024).

Using a fixed LLM and a shared evaluation protocol, we first quantify the effect of data alone by fine-tuning on MEGA versus other public datasets. We then show that post-training with GRPO Shao et al. (2024), using a composite reward that combines a thresholded property gain term and a Tanimoto similarity term Bajusz et al. (2015), yields further gains with reduced number of training samples.

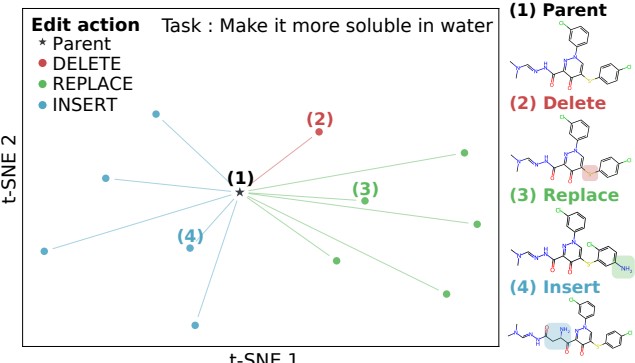

Figure 1: Morgan-fingerprint t-SNE for a parent SMILES and child molecules generated by fragment edits, delete, replace, insert. Colors encode the applied edit, highlighting neighborhood exploration under the given task.

Concretely, this work introduces the following contributions:

1. We release MEGA, a family of molecular editing datasets with fragment-level Replace, Insert, and Delete annotations. It contains two variants: MEGA (522K pairs) for resource-constrained regimes, and MEGA-Large (31M positive and 41M negative pairs) for scaling and contrastive studies. MEGA-Large is over an order of magnitude larger than any existing dataset for molecular editing.

2. We demonstrate that under fixed model and training protocol, fine-tuning on MEGA increases hit ratios by up to +21.47 percentage points over established datasets on shared tasks, while its explicit edit labels enable per-action supervision and diagnostics.

3. We show that GRPO post-training on MEGA with a similarity-aware reward improves property alignment and edit minimality, while also achieving state-of-the-art performance on established benchmarks. With only 14K training examples, the GRPO post-trained model matches the performance of the SFT model trained on the full MEGA set, corresponding to a $\sim 36\times$ improvement in data efficiency.

## 2 RELATED WORK

### 2.1 DATASETS FOR MOLECULAR EDITING

Public corpora vary in task formulation and scale. MoleculeSTM Liu et al. (2023a) trains a multimodal structure–text model on hundreds of thousands of molecule–caption pairs through contrastive learning and proposes instruction-guided retrieval and editing tasks, establishing a text-based benchmark for property-aware modification. Another example is MolOpt-Instructions Ye et al. (2025), released alongside DrugAssist, which compiles a large instruction dataset to fine-tune language models for molecule optimization from natural language goals. Furthermore, MolEdit-Instruct Zhuang et al. (2025) scales property-conditioned edits by pairing each parent molecule with an explicit edit instruction and target property change. The dataset is used to evaluate diffusion and RL models under joint constraints on molecular similarity and property improvement, reflecting a shift toward instruction-plus-constraint benchmarks. Together, these datasets illustrate the available range for training and evaluating molecular editing models, despite differences in construction, supervision signals, and scale.

### 2.2 LLMS FOR CHEMISTRY

General-purpose language models trained on broad text data already exhibit useful zero-shot chemistry skills answering property prediction questions, translating line notations, or suggesting

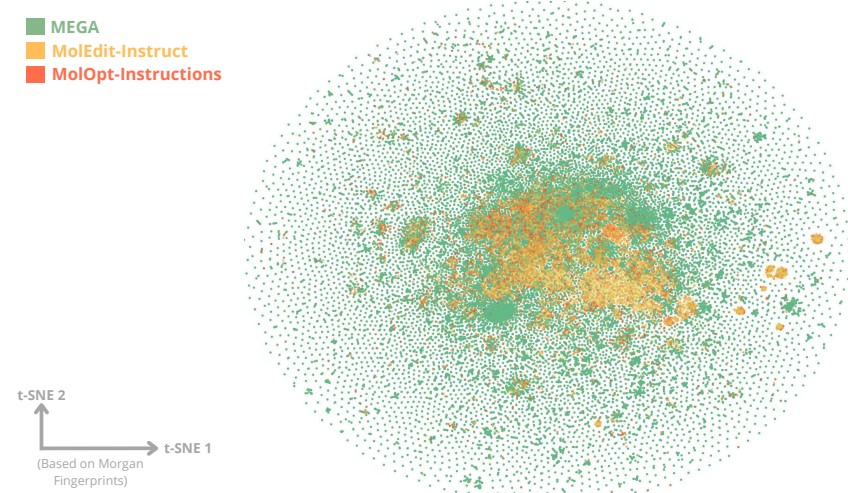

Figure 2: t-SNE projection of Morgan fingerprints showing chemical space coverage of statistical significant subsets of MEGA, MolEdit-Instruct, and MolOpt-Instructions.

functional-group swaps straight out of the box Liu et al. (2023b); Bran et al. (2025); Mirza et al. (2025b). When wrapped in a tool-calling framework, the same models can act as agents: ChemCrow, for example, prompts an off-the-shelf LLM to invoke cheminformatics utilities (parsers, property predictors, similarity search) and carry out multi-step design tasks from natural language instructions M. Bran et al. (2024).

Researchers also adapt these open language models to chemistry via domain fine-tuning and task-specific supervision. For instance, LlamoLe trains on ∼128k USPTO reactions with textual descriptions to strengthen reasoning and route identification Lowe (2017); Liu et al. (2024), while DrugAssist uses MolOpt-Instructions to instruction-tune models for property-directed optimization from text in a single-shot fashion Ye et al. (2025).

A further layer of refinement uses reinforcement learning such as with Ether0, trained on 640k experimentally-grounded chemistry problems across 375 tasks, to excel at tasks like retrosynthesis and solubility editing Narayanan et al. (2025). Another example is MolEditRL, which pairs property-conditioned prompts with structure-preserving edit operators and reinforcement-style objectives to promote local, similarity-respecting modifications Zhuang et al. (2025).

## 2.3 EDITORS BEYOND LLMS FOR LEAD OPTIMIZATION

While LLM-based editors are comparatively recent, lead optimization has a long history of non-LLM approaches that emphasize local, property-directed modifications to a given scaffold. Earlier rule-based strategies, such as matched molecular pairs (MMPs) Yang et al. (2023) and fixed reaction templates, encoded medicinal-chemistry heuristics for systematic substitution. More recent machine learning methods operate directly on strings or graphs to propose minimal edits, including JT-VAE Jin et al. (2019), GCPN You et al. (2019), and MARS Xie et al. (2021); Loeffler et al. (2024). In parallel, diffusion models adapt continuous generative dynamics to discrete molecular modifications: DiffLink designs linkers between fixed fragments Igashov et al. (2024), while DiffHop performs constrained scaffold hopping Torge et al. (2023). Taken together, these approaches chart a progression from rules to learned editors to diffusion frameworks, all aimed at controllable, chemically plausible edits central to lead optimization.

## 3 MEGA DATASET

### 3.1 DATASET CONSTRUCTION OVERVIEW

MEGA is a family of large-scale datasets with a total of 31.4M parent–child SMILES pairs. Each child comes from applying a single functional-group edit to a ZINC250K parent, without a constraint to preserve the scaffold Irwin & Shoichet (2005). Candidate modification sites are located with established retrosynthetic slicing rules (BRICS Degen et al. (2008), Hussain–Rea (HR) Hussain & Rea (2010) and RECAP Lewell et al. (1998)) and exactly one action is applied at a chosen site: *Delete*, *Insert*, or *Replace* a functional group. The child is rebuilt and sanitized in RDKit Landrum & Contributors (2025), and task properties are computed deterministically. We adopt the MoleculeSTM protocol for task labeling: for each objective (e.g. "increase solubility"), we use RDKit to verify whether the child clears the threshold for that task. Each record includes parent SMILES, child SMILES, a coarse action tag (*Insert/Delete/Replace*), the task identifier and threshold level, and the parent/child property vectors. The computational budget for MEGA amounted to approximately 184k CPU-hours on a 128-core cluster.

For efficient training, we define MEGA as a uniformly sampled subset of 522K positive examples drawn from the full 31M-pair dataset, which we refer to as MEGA-Large. MEGA mirrors the action distribution of MEGA-Large (3.1% *Delete*, 43.6% *Insert*, 53.3% *Replace*), making it suitable for resource-constrained settings while retaining the statistical properties of the full collection. In addition to the positives, we also provide 41 million valid but chemically close negative pairs that, while not meeting the improvement threshold, offer valuable hard negatives for contrastive learning or reinforcement learning setups.

To emphasize drug discovery relevance, our tasks target widely used objectives including aqueous solubility, drug-likeness (QED), H-bond donors/acceptors, permeability proxies and topological polar surface area (TPSA). Each one evaluated at two thresholds, loose and strict. Restricting edits to a single modification per pair enables controlled exploration of the parent's local chemical neighborhood. A parent molecule may appear in multiple pairs if it contains eligible sites for several actions across tasks. For each edit–task combination, we retain up to five successful and five near-miss children, ranked to maximize diversity while avoiding redundancy. Further details on tasks and dataset composition are provided in Appendix A.

### 3.2 DATASET COVERAGE

Figure 1 shows a representative parent alongside three children, one per action. The edits are local and chemically rational: removing an atom (*Delete*), adding a small moiety (*Insert*) or swapping one group for another (*Replace*). Together they illustrate the targeted nature of MEGA's pairs. In this example, all children satisfy the "increase aqueous solubility" objective.

Figure 2 visualizes a statistically significant subset of MEGA in the 2048-bit Morgan-fingerprint space Morgan (1965) using t-SNE van der Maaten & Hinton (2008). The overlay shows that MEGA occupies the shared high-density core with existing molecular editing datasets and also reaches beyond it, consistent with its scale and edit policy. Moreover, Table 1 quantifies this comparison: in terms of successful (positive) edits, the full set is roughly an order of magnitude larger than the next largest dataset. Furthermore, unlike other datasets, MEGA includes a coarse action label (*Insert/Delete/Replace*) for every pair, supporting per-action supervision, diagnostics, and reproducibility.

## 4 EXPERIMENTS

We evaluate MEGA in a two-stage protocol: (1) supervised fine-tuning (SFT) to benchmark performance under identical model and training settings against existing datasets, and (2) RL post-training with a hybrid reward combining property gains and structural similarity. We also analyze edit action distributions, locality, and sample efficiency in single- and multi-objective tasks.

Table 1: Comparison of molecular editing datasets used in this study. Reported sizes count only successful (positive) parent-child pairs. Unique molecules counts distinct SMILES across both parents and children. Action provided indicates whether a dataset records the edit label.

| Dataset | Size | Unique Molecules | # Tasks | Action Provided |
|---|---|---|---|---|
| MoleculeSTM | 280K | 250K | 34 | No |
| MolEdit-Instruct | 3.03M | 967K | 20 | No |
| MolOpt-Instructions | 1.24M | 1.596M | 16 | No |
| MEGA | 522K | 372K | 28 | Yes |
| MEGA-Large | 31.4M | 22.126M | 28 | Yes |

Table 2: Performance comparison of SFT models on shared molecular editing tasks. The best results are marked in bold. We report the mean and std of five runs.

| Task | Description | Threshold | Dataset | | |
|---|---|---|---|---|---|
| | | | **MEGA** | **MolEdit Instruct** | **MolOpt Instruction** |
| 103 | *More like a drug* | 0.0 | **62.46 ± 2.18** | 23.92 ± 0.99 | 16.38 ± 2.03 |
| | | 0.1 | **28.43 ± 1.38** | 12.85 ± 0.58 | 8.38 ± 0.53 |
| 104 | *Less like a drug* | 0.0 | 97.81 ± 0.91 | **98.97 ± 0.33** | 96.87 ± 0.82 |
| | | 0.1 | 83.94 ± 3.43 | **98.86 ± 0.51** | 94.43 ± 1.47 |
| 107 | *More H-bond acceptors* | 0.0 | **99.28 ± 0.25** | 94.96 ± 1.70 | 89.33 ± 1.18 |
| | | 1.0 | **93.06 ± 0.66** | 43.35 ± 1.65 | 34.06 ± 0.58 |
| 108 | *More H-bond donors* | 0.0 | **99.80 ± 0.25** | 97.66 ± 0.59 | 96.21 ± 0.91 |
| | | 1.0 | **99.29 ± 0.25** | 67.57 ± 1.78 | 56.67 ± 1.10 |
| | **Average** | | **83.01** | 67.27 | 61.54 |

## 4.1 SUPERVISED FINE-TUNING

**Protocol.** We fine-tune a Llama-3 8B model Dubey et al. (2024) with LoRA adapters Hu et al. (2021) on MolOpt Instruction, MolEdit Instruct, and MEGA. All runs use the same hyperparameters, training schedule, and LoRA configuration. Training last approximately 23 H100-equivalent hours per model until the validation loss no longer improves. For further training details see Appendix D.1.

Evaluation follows the MoleculeSTM protocol Liu et al. (2023a) and is restricted to the 4 single-objective tasks shared by all three datasets. The test set contains 200 unique parent SMILES not present in any of the training sets. For each task, we assess performance at two property thresholds (loose and strict) and report the hit ratio, defined as the fraction of generated molecules that achieve the required property improvement. Each experiment is repeated five times, with a decoding temperature of 1.0, and we report the mean and standard deviation of the hit ratio across runs. The resulting models are referred to as MEGA SFT.

**Results.** Table 2 shows that the LLM trained on MEGA outperforms the same architecture trained on MolEdit Instruct and MolOpt Instruction by +15.74 (pp) and +21.47 (pp), respectively. The largest gain occurs in the "more like a drug" objective, a target known to be particularly challenging due to its composite nature Liu et al. (2023b). Variance is low and comparable to the other benchmarks, indicating that improvements are stable across repeated evaluations. For further comparisons refer to Appendix C.2.

## 4.2 REWARD-GUIDED POST-TRAINING

**Protocol.** We further refine the best checkpoint from above using GRPO Shao et al. (2024) to improve property alignment while preserving local edits. During training, for each parent SMILES,

Table 3: Comparison of DrugAssist, Gemini 2.5 Pro, and MEGA-GRPO across five shared tasks from the DrugAssist benchmark, evaluated under loose and strict thresholds.

| Task | Description | Threshold | Model | | |
|------|-------------|-----------|-----------|----------------|-----------|
| | | | **DrugAssist** | **Gemini 2.5 Pro** | **MEGA GRPO** |
| 101 | *More soluble in water* | 0 | 80.00 | 82.23 | **97.49** |
| | | 0.5 | 41.00 | 59.45 | **91.10** |
| 103 | *More like a drug* | 0 | 76.00 | 60.14 | **83.49** |
| | | 0.1 | **63.00** | 23.46 | 50.00 |
| 107 | *More H-bond acceptors* | 0 | 71.00 | 64.97 | **98.60** |
| | | 1 | 67.00 | 5.57 | **86.74** |
| 108 | *More H-bond donors* | 0 | 72.00 | 73.54 | **99.31** |
| | | 1 | 76.00 | 6.32 | **91.45** |
| 201 | *More soluble & more HBA* | 0 - 0 | 50.00 | 80.32 | **95.19** |
| | | 0.5 - 1 | 27.00 | 24.43 | **84.21** |
| | **Average** | | 62.30 | 48.05 | **87.76** |

the model generates a batch of multiple candidates, which are scored relative to each other. This feedback is used for updating the model weights. The scalar reward is defined as:

$$R = \underbrace{\mu \cdot \mathbb{1}[\Delta p(\text{parent}, \text{child}) \geq \tau]}_{\text{property hit}} + \gamma \underbrace{\mathbb{1}_{\text{valid}}(\text{child})}_{\text{validity hit}} + \lambda \underbrace{h_{\text{tan}}(\text{parent}, \text{child})}_{\text{Tanimoto hit level}}$$

$$h_{\text{tan}}(\text{parent}, \text{child}) = \begin{cases} 1.0, & \text{if } T > 0.65, \\ 0.5, & \text{if } 0.4 \leq T \leq 0.65, \\ 0.0, & \text{otherwise}, \end{cases}$$

where the first term awards a hit when $\Delta p$ meets or exceeds $\tau$, with $\mu = 0.5$ for satisfying the task without margin (loose threshold) and $\mu = 1.0$ for satisfying it with a strict margin (strict threshold). The second term rewards valid and sanitized child SMILES, and the third rewards scaffold-local modifications via Tanimoto coefficient discretization. The coefficients $\gamma$ and $\lambda$ were selected empirically to 1.0. Importantly, the property hit reward function depends only on the property oracle output $\Delta p$, making it independent of SFT annotations. This allows MEGA-trained models to adapt to new endpoints for which a property oracle exists, without requiring newly annotated training pairs. Full details are in Appendix D.2.

To assess data efficiency, we train models on subsets ranging from 1.4k to 522k examples (MEGA GRPO). We first compare MEGA GRPO against DrugAssist Zhuang et al. (2025), a state-of-the-art specialized LLM, and Gemini 2.5 Pro Comanici et al. (2025), a strong general-purpose LLM, on five single- and multi-objective molecular editing tasks. For this evaluation, we use the 500 SMILES test set provided by DrugAssist and report hit ratios under both loose and strict thresholds in Table 3.

We then compare MEGA GRPO against ChatDrug Turbo, a strong in-context learning LLM, and MoleculeSTM, a contrastive-trained encoder–decoder, on the full 28-task suite of the MEGA dataset. For this evaluation, we follow the same protocol described in the SFT section and report results in Table 4.

**Results.** MEGA GRPO outperforms both DrugAssist and Gemini 2.5 Pro on the DrugAssist benchmark (Table 3), achieving the highest hit ratio in 9 of 10 settings. The most pronounced gains appear on the dual-objective solubility + HBA task (201), where it reaches 95.19% under loose and 84.21% under strict thresholds, substantially ahead of both baselines. The only case where MEGA GRPO underperforms is the strict drug-likeness objective, where DrugAssist retains an edge. Gemini 2.5 Pro consistently trails, particularly under strict thresholds, underscoring the difficulty of zero-shot general-purpose LLMs in molecular editing.

Table 4: Performance comparison of MEGA GRPO (522K) against editing methods across single and multi objective tasks and thresholds. We report the mean and standard deviation over five runs. The best results are shown in bold.

| Task | Threshold | Random | MoleculeSTM | ChatDrug Turbo | MEGA GRPO |
|------|-----------|--------|-------------|----------------|-----------|
| 101 | 0 | 35.33±1.31 | 61.87±2.67 | 94.13±1.04 | **99.31±0.10** |
|     | 0.5 | 11.04±2.40 | 49.02±1.84 | 88.67±0.95 | **94.43±0.24** |
| 102 | 0 | 43.36±3.06 | 52.71±1.67 | 96.86±1.10 | **99.71±0.21** |
|     | 0.5 | 19.75±1.56 | 30.47±3.26 | 70.08±3.44 | **95.52±0.51** |
| 103 | 0 | 38.06±2.57 | 36.52±2.46 | 48.65±3.39 | **60.48±2.28** |
|     | 0.1 | 5.27±0.24 | 8.81±0.82 | 19.37±5.54 | **23.38±1.71** |
| 104 | 0 | 36.96±2.25 | 58.59±1.01 | 70.75±2.92 | **97.81±0.73** |
|     | 0.1 | 6.16±1.87 | 37.56±1.76 | 30.99±2.66 | **93.42±0.58** |
| 105 | 0 | 25.23±2.13 | 57.74±0.60 | 56.56±1.84 | **90.19±1.34** |
|     | 10 | 17.41±1.43 | 47.51±1.88 | 43.08±2.95 | **87.88±0.94** |
| 106 | 0 | 16.79±2.54 | 34.13±0.59 | 77.35±1.98 | **100.00±0.00** |
|     | 10 | 11.02±0.71 | 26.48±0.97 | 66.69±2.74 | **99.43±0.01** |
| 107 | 0 | 12.64±1.64 | 54.01±5.26 | 95.35±0.62 | **99.86±0.29** |
|     | 1 | 0.69±0.01 | 27.33±2.62 | 72.60±2.51 | **88.12±1.27** |
| 108 | 0 | 2.97±0.61 | 28.55±0.76 | 96.54±1.31 | **99.24±0.79** |
|     | 1 | 0.00±0.00 | 7.69±0.56 | 76.43±3.32 | **95.22±0.34** |
| 201 | 0 − 0 | 9.88±1.03 | 27.87±3.86 | 79.62±0.64 | **98.53±0.44** |
|     | 0.5 − 1 | 0.23±0.33 | 8.80±0.04 | 49.64±2.66 | **90.34±0.47** |
| 202 | 0 − 0 | 2.99±0.38 | 8.55±2.75 | 51.59±3.79 | **94.18±1.20** |
|     | 0.5 − 1 | 0.45±0.32 | 2.93±0.30 | 24.92±4.85 | **72.59±1.41** |
| 203 | 0 − 0 | 2.28±1.15 | 33.51±4.08 | 89.34±0.96 | **99.64±0.48** |
|     | 0.5 − 1 | 0.00±0.00 | 9.98±1.03 | 53.64±5.81 | **98.35±0.90** |
| 204 | 0 − 0 | 0.69±0.58 | 17.03±2.75 | 39.90±3.86 | **86.72±1.95** |
|     | 0.5 − 1 | 0.00±0.00 | 2.59±1.14 | 24.19±2.19 | **60.33±1.83** |
| 205 | 0 − 0 | 5.06±1.21 | 35.69±3.19 | 12.85±2.68 | **89.30±0.93** |
|     | 0.5 − 10 | 1.16±0.68 | 19.15±0.73 | 10.44±5.75 | **82.86±0.75** |
| 206 | 0 − 0 | 12.17±1.05 | 44.35±0.68 | 65.33±2.16 | **99.54±0.43** |
|     | 0.5 − 10 | 6.20±0.64 | 28.67±2.22 | 52.90±2.23 | **94.31±0.23** |

On the 28-task MoleculeSTM benchmark (Table 4), MEGA GRPO attains the best mean hit ratio on all task/threshold pairs. It reaches ≥95% on most single-property edits under loose thresholds (e.g., 101–102, 104, 106–108) and remains strong under stricter criteria. MEGA GRPO's advantage is most pronounced on multi-objective tasks (201, 203, and 206), indicating better balancing of potentially competing constraints. For further comparison with other benchmarks see Appendix C.1.

**Data Efficiency.** Figure 3 shows that GRPO with Tanimoto reward outperforms SFT across all data regimes while maintaining scaffold edits within our targeted Tanimoto similarity range (0.6–0.8). With only 14k training examples, MEGA GRPO (14K) matches the performance of MEGA SFT trained on 522k by +2.11 pp, achieving $\sim 36\times$ data efficiency multiplier with the same Llama 3 base model.

**Generalization to Unseen Properties.** We test whether MEGA-GRPO can generalize beyond its training distribution. On three biological endpoints never seen during training (DRD2, JNK3, GSK3$\beta$), the model achieves 83.0% mean success rate compared to 7.0% for the base model (+76.0pp). On three held-out multi-objective tasks (LogP↑+TPSA↓, QED↑+TPSA↑,

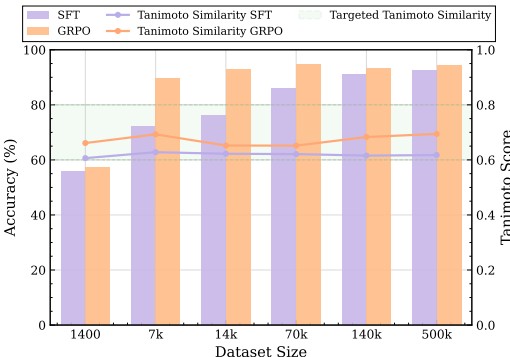

Figure 3: Data efficiency comparison of SFT and GRPO across training set sizes (based on loose threshold). GRPO consistently outperforms SFT while keeping edits within the targeted Tanimoto similarity range (0.6–0.8).

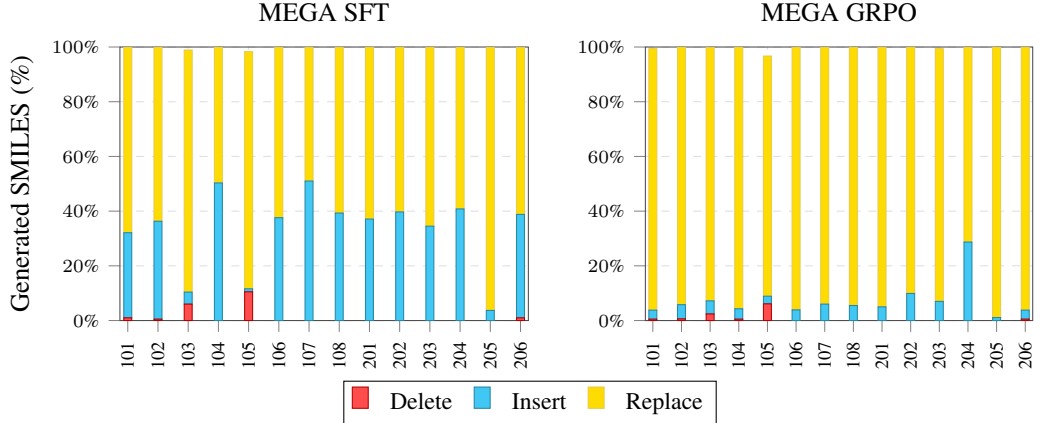

Figure 4: Distribution of fragment-level edit actions during inference for MEGA SFT (522K) and MEGA GRPO (14K), on single and double-molecule optimization tasks.

HBA↑+TPSA↑), it reaches 90.3% success rate and 54.7% hit rate versus 5.5% and 2.2% for the base model. These results demonstrate that the model learns molecular editing skills that extend beyond its training corpus. Full evaluation details are in Appendix B.

**Guided-Action Editing.** Figure 4 shows the distribution of fragment-level edit actions across tasks. The MEGA SFT model roughly reproduces the action distribution of the MEGA dataset, indicating internalization of single-fragment edit patterns present in the demonstrations. In contrast, MEGA GRPO, trained via RL, favors *replace* actions, reflecting an optimization bias towards minimal yet property-aligned functional group modifications. For more details on the model's behavior, see Appendix E.

## 5 CONCLUSION

In this work, we release MEGA, a family of large-scale datasets comprising 72M molecule pairs designed to advance property-guided molecular editing. By systematically generating single chemically rational edits that improve a target property (*replace*, *insert*, *delete*), MEGA provides dense, high-quality supervision for exploring local chemical space. Our experiments demonstrate its value: models fine-tuned on MEGA significantly outperform those trained on existing datasets in supervised settings. Furthermore, when combined with RL post-training, models trained on MEGA achieve state-of-the-art performance on established benchmarks, demonstrate a remarkable $\sim 36\times$ improvement in data efficiency, and exhibits generalization to unseeing properties and tasks.

## 6 Reproducibility Statement

The construction of the dataset is described in Section 3.1, with additional licensing details provided in Appendix F. The dataset is released under the same license as the source dataset from which it is derived. Scripts for training and evaluation, including both fine-tuning and post-training procedures, will be made available in a public repository under a permissive license. These resources, along with the detailed experimental settings reported in the main text and appendix, are intended to ensure full reproducibility of our results and maximize benefit to the research community.

## 7 Ethical Statement

The authors have considered the ethical implications of this work and find no direct ethical concerns. All research was performed on publicly available datasets and did not involve human subjects or personally identifiable information. The authors also declare no conflicts of interest.

## 8 Use of Large Language Models

Large Language Models (LLMs) were used in a limited capacity as a writing and coding assistant. Specifically, LLMs were used to:

- Improve clarity, grammar, and flow in parts of the manuscript.
- Suggest structure for some sections of text.
- Assist with debugging of boilerplate code, without contributing novel algorithmic ideas.

All substantive research ideas, experimental design, analysis, and conclusions were developed entirely by the authors. The authors take full responsibility for the accuracy and integrity of all content presented.

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

# A MEGA DATASET DETAILS

## A.1 DATASET FORMAT AND TASKS

**Structure and Format.** MEGA is released in two versions: MEGA (522K successful edits, 522K negative edits) and MEGA-Large (31.4M successful edits, 41M negative edits), both derived from exhaustive exploration of the ZINC250 chemical space. Each dataset record contains 11 fields capturing the complete molecular editing task: `task_id` (int64) identifies one of 14 optimization objectives (Tasks 101–108 for single-objective, 201–206 for multi-objective); `prompt` (string) provides the natural-language instruction; `input_smiles` and `output_smiles` (strings) represent the parent and child molecules in SMILES format; `action_type` (categorical: insert, delete, replace) specifies the edit operation. Critically, the `edit` field provides natural-language action-wise tracing of the exact structural modification, explicitly describing which fragment was replaced and with what, which fragment was removed, or the neighborhood context where an insertion occurred—enabling transparent interpretation of chemical transformations. `target_delta` (float64) quantifies the property change magnitude, while four additional descriptors (`SA_delta`, `MW_delta`, `QED_delta`, `murcko_scaffold_retained`) characterize trade-offs and structural conservation. Both datasets are partitioned into 90/10 train/validation splits, with separate files for successful (positive) and unsuccessful (negative) edits to support contrastive learning and reinforcement learning applications. The consistent schema across scales enables seamless scaling experiments and reproducible benchmarking.

**Tasks.** For curating MEGA we used single-objective tasks (101–108) that targets one property, and multi-objective tasks (201–206) for two properties. Table 5 lists the desired direction of change ($\uparrow$ increase, $\downarrow$ decrease), variable name (consistent with RDKit), alongside the requirement in natural-language. For each task we evaluate 2 threshold with different levels of property change. Table 6 gives the evaluation thresholds under *loose* and *strict* criteria. For multi-objective tasks, each threshold vector follows the property order in the *Target(s)* column.

| Task ID | Target(s) | Task Requirement 1 | Task Requirement 2 |
|---|---|---|---|
| 101 | $\downarrow \log P$ | more soluble in water | None |
| 102 | $\uparrow \log P$ | less soluble in water | None |
| 103 | $\uparrow$ QED | more like a drug | None |
| 104 | $\downarrow$ QED | less like a drug | None |
| 105 | $\downarrow$ TPSA | higher permeability | None |
| 106 | $\uparrow$ TPSA | lower permeability | None |
| 107 | $\uparrow$ HBA | more hydrogen bond acceptors | None |
| 108 | $\uparrow$ HBD | more hydrogen bond donors | None |
| 201 | $\downarrow \log P, \uparrow$ HBA | more soluble in water | more hydrogen bond acceptors |
| 202 | $\uparrow \log P, \uparrow$ HBA | less soluble in water | more hydrogen bond acceptors |
| 203 | $\downarrow \log P, \uparrow$ HBD | more soluble in water | more hydrogen bond donors |
| 204 | $\uparrow \log P, \uparrow$ HBD | less soluble in water | more hydrogen bond donors |
| 205 | $\downarrow \log P, \downarrow$ TPSA | more soluble in water | higher permeability |
| 206 | $\downarrow \log P, \uparrow$ TPSA | more soluble in water | lower permeability |

Table 5: Task catalog for small-molecule property edits. All tasks require the output molecule to remain similar to the input. Arrows indicate desired property direction.

## A.2 DATASET STATISTICS

This subsection summarizes the scale and composition of MEGA-Large (31M) and MEGA (522K) and quantifies how representative the smaller split is of the full corpus. Table 7 reports dataset-level counts. MEGA-31M contains 246,532 unique parent molecules directly taken from the Zinc-250 dataset. In includes 72,366,584 evaluated edits, of which 31,354,522 are successful. MEGA mirrors this profile at smaller scale with 4,105 unique parents and 1,205,430 edits, including 522,058.

| Task ID | Loose | Strict |
|---------|-------|--------|
| 101 | [0] | [0.5] |
| 102 | [0] | [0.5] |
| 103 | [0] | [0.1] |
| 104 | [0] | [0.1] |
| 105 | [0] | [10] |
| 106 | [0] | [10] |
| 107 | [0] | [1] |
| 108 | [0] | [1] |
| 201 | [0, 0] | [0.5, 1] |
| 202 | [0, 0] | [0.5, 1] |
| 203 | [0, 0] | [0.5, 1] |
| 204 | [0, 0] | [0.5, 1] |
| 205 | [0, 0] | [0.5, 10] |
| 206 | [0, 0] | [0.5, 10] |

Table 6: Evaluation thresholds per task. For multi-objective tasks, each vector's order follows the *Target(s)* order in Table 5.

| Metric | MEGA-Large (31M) | MEGA (522K) |
|--------|------------------|-------------|
| Unique parent molecules | 246,532 | 4,105 |
| Successful edits | 31,354,522 | 522,058 |
| Unique successful SMILES | 21,879,431 | 367,954 |
| Negative edits | 41,012,062 | 683,372 |
| Unique negative SMILES | 8,129,138 | 137,012 |
| Total SMILES | 72,366,584 | 1,205,430 |

Table 7: Side-by-side summary of MEGA datasets.

**Duplication Statistics.** Table 8 reports duplication rates in MEGA at the parent and child molecule levels. Parent molecules exhibit 99.2% reuse across the dataset, reflecting the exhaustive exploration of edit operations applied to each ZINC250 molecule. In contrast, child molecules show a mean duplication rate of 30.2%, resulting from convergent edits where different parent molecules or edit operations produce identical products for several tasks. This convergence is chemically meaningful and indicates that multiple structural modifications can lead to the same optimized structure. The substantial difference between parent and child duplication rates confirms that MEGA captures diverse optimization pathways while maintaining structural diversity in the output space.

Table 8: Duplication statistics for parent and child molecules in MEGA.

| Molecule Type | Duplication Rate | Description |
|---------------|------------------|-------------|
| Parent molecules | 99.2% | ZINC250 original SMILES reused across tasks |
| Child molecules | 30.2% | Mean duplication from convergent edits |

**Per-Task Exploration Efficiency.** Table 9 reports exploration efficiency as the average number of edits exceeding task-wise strict thresholds per exploration episode, stratified by action type. These metrics are computed over 10.4M exploration episodes (249,455 per task-action combination) spanning 11.3B oracle calls to RDKit (NFE, Number of Function Evaluations). Lower efficiency values indicate greater search effort required to discover successful edits, serving as a natural proxy for task-action difficulty. Delete operations consistently show the lowest efficiency across tasks, while Insert and Replace operations achieve higher success rates. Multi-objective tasks (201–206) and QED optimization (103) exhibit particularly challenging search landscapes, requiring substantially more computational effort to identify property-improving edits.

Table 10 compares the distribution of successful edits by operation. The proportions are stable across scales: delete ≈3.1%, insert ≈43.6%, and replace ≈53.3% in both MEGA (522K)

Table 9: Per-task exploration efficiency by action type. Values show average successful edits per episode; NFE in parentheses indicates total oracle calls.

| Task | Objective | DELETE | INSERT | REPLACE |
|------|-----------|--------|--------|---------|
| 101 | LogP↓ | 0.61 (753K) | 4.93 (15.0M) | 4.94 (10.9M) |
| 102 | LogP↑ | 0.59 (753K) | 4.93 (9.0M) | 4.94 (8.7M) |
| 103 | QED↑ | 0.37 (753K) | 0.61 (1,798M) | 3.27 (2,013M) |
| 104 | QED↓ | 0.43 (753K) | 4.93 (8.1M) | 4.94 (8.3M) |
| 105 | TPSA↓ | 1.70 (751K) | 0.00 (1,923M) | 4.89 (759M) |
| 106 | TPSA↑ | 0.00 (753K) | 4.93 (7.6M) | 4.94 (5.6M) |
| 107 | HBA↑ | 0.00 (753K) | 4.93 (7.3M) | 4.94 (14.4M) |
| 108 | HBD↑ | 0.00 (753K) | 4.93 (23.5M) | 4.94 (44.2M) |
| 201 | LogP↓+HBA↑ | 0.00 (753K) | 4.93 (15.7M) | 4.94 (47.4M) |
| 202 | LogP↑+HBA↑ | 0.00 (753K) | 4.93 (11.6M) | 4.92 (150.3M) |
| 203 | LogP↓+HBD↑ | 0.00 (753K) | 4.93 (29.2M) | 4.94 (55.3M) |
| 204 | LogP↑+HBD↑ | 0.00 (753K) | 4.90 (279.0M) | 4.73 (923.8M) |
| 205 | LogP↓+TPSA↓ | 0.15 (753K) | 0.00 (1,923M) | 4.73 (1,214M) |
| 206 | LogP↓+TPSA↑ | 0.00 (753K) | 4.93 (15.1M) | 4.94 (14.8M) |
| **Overall** | **(all)** | **0.28 (10.5M)** | **3.92 (6.1B)** | **4.79 (5.3B)** |

and MEGA-Large (31M). This alignment suggests that MEGA preserves the operational mix of the full dataset and is suitable for compute-friendly budgets.

| | MEGA (522K) | | MEGA-Large (31M) | |
|-----------|------|------|------------|------|
| **Operation** | **Count** | **%** | **Count** | **%** |
| Delete | 15,924 | 3.1% | 960,992 | 3.1% |
| Insert | 227,789 | 43.6% | 13,677,420 | 43.6% |
| Replace | 278,345 | 53.3% | 16,716,110 | 53.3% |
| **Total** | 522,058 | 100% | 31,354,522 | 100% |

Table 10: Distribution of successful edit operations for MEGA-Large and MEGA.

**Per-Task Action Distribution.** Table 11 reports the distribution of successful edit operations by action type across all tasks in MEGA. Action distributions vary significantly by optimization objective, reflecting task-specific structural requirements. Solubility tasks (101, 102) exhibit balanced Insert/Replace distributions ( 50/50), while other tasks show more skewed patterns. Task 103 (QED↑) favors Replace operations (77%), Task 105 (TPSA↓) excludes Insert operations entirely (0%), and Task 205 (LogP↓+TPSA↓) is dominated by Replace operations (97%). These variations demonstrate that MEGA captures task-dependent edit strategies aligned with chemical intuition.

Table 12 reports successful edits per task for MEGA-Large (31M) and MEGA (522K). Counts are broadly balanced across tasks and per-task ranking is consistent across scales. Tasks 101/102/104 yield the largest winner pools, while 103 (increase QED) and 205 (reduce $\log P$ & decrease TPSA) show markedly consistent with results from the literature. MEGA preserves the relative task difficulty profile of the full corpus.

A.3    CHEMICAL TRANSFORMATION CHARACTERISTICS

**Chemical Diversity Per Task.** Table 13 reports scaffold diversity computed via Bemis-Murcko decomposition on 50,000 stratified random samples (95% confidence interval, ±1% margin). Diversity is measured as the ratio of unique scaffolds to total molecules within each task's successful edits. Per-task scaffold diversity ranges from 0.91 to 0.98, indicating substantial structural variation within individual optimization objectives. The overall dataset diversity of 0.88 confirms that MEGA spans a broad chemical space while maintaining task-specific structural coherence. These high diversity values demonstrate that the dataset avoids trivial repetition and provides models with exposure to varied molecular architectures during training.

Table 11: Distribution of successful edit operations by task. Proportions reflect task-specific structural requirements for property optimization.

| Task | Objective | Delete | % | Insert | % | Replace | % |
|------|-----------|--------|-----|--------|------|---------|------|
| 101 | LogP↓ | 1,294 | 3.0% | 21,123 | 48.6% | 21,046 | 48.4% |
| 102 | LogP↑ | 1,287 | 3.0% | 21,095 | 48.6% | 21,061 | 48.5% |
| 103 | QED↑ | 658 | 3.7% | 3,429 | 19.3% | 13,687 | 77.0% |
| 104 | QED↓ | 1,286 | 3.0% | 20,871 | 48.8% | 20,636 | 48.2% |
| 105 | TPSA↓ | 4,660 | 17.0% | 0 | 0.0% | 22,741 | 83.0% |
| 106 | TPSA↑ | 0 | 0.0% | 20,502 | 50.0% | 20,503 | 50.0% |
| 107 | HBA↑ | 0 | 0.0% | 20,502 | 50.0% | 20,503 | 50.0% |
| 108 | HBD↑ | 0 | 0.0% | 20,502 | 50.0% | 20,503 | 50.0% |
| 201 | LogP↓+HBA↑ | 0 | 0.0% | 20,502 | 50.0% | 20,503 | 50.0% |
| 202 | LogP↑+HBA↑ | 0 | 0.0% | 20,466 | 50.0% | 20,467 | 50.0% |
| 203 | LogP↓+HBD↑ | 0 | 0.0% | 20,502 | 50.0% | 20,503 | 50.0% |
| 204 | LogP↑+HBD↑ | 0 | 0.0% | 19,989 | 50.0% | 19,989 | 50.0% |
| 205 | LogP↓+TPSA↓ | 609 | 3.0% | 0 | 0.0% | 19,634 | 97.0% |
| 206 | LogP↓+TPSA↑ | 0 | 0.0% | 20,502 | 50.0% | 20,503 | 50.0% |
| **Total** | | **9,794** | **1.9%** | **229,985** | **44.0%** | **282,279** | **54.1%** |

| Task | MEGA-Large | MEGA |
|------|------------|------|
| 101 | 2,613,794 | 43,463 |
| 102 | 2,609,126 | 43,443 |
| 103 | 1,061,168 | 17,774 |
| 104 | 2,570,496 | 42,793 |
| 105 | 1,645,706 | 27,401 |
| 106 | 2,462,800 | 41,005 |
| 107 | 2,462,791 | 41,005 |
| 108 | 2,462,781 | 41,005 |
| 201 | 2,462,711 | 41,005 |
| 202 | 2,457,965 | 40,933 |
| 203 | 2,462,768 | 41,005 |
| 204 | 2,400,936 | 39,978 |
| 205 | 1,218,686 | 20,243 |
| 206 | 2,462,794 | 41,005 |
| **Total** | **31,354,522** | **522,058** |

Table 12: Number of successful edit examples per task for MEGA-Large (31M) and MEGA (522K).

Table 13: Scaffold diversity per task computed via Bemis-Murcko decomposition on 50K stratified samples.

| Task | Objective | Scaffold Diversity |
|------|-----------|--------------------|
| 101 | LogP↓ | 0.94 |
| 102 | LogP↑ | 0.93 |
| 103 | QED↑ | 0.96 |
| 104 | QED↓ | 0.91 |
| 105 | TPSA↓ | 0.95 |
| 106 | TPSA↑ | 0.92 |
| 107 | HBA↑ | 0.97 |
| 108 | HBD↑ | 0.98 |
| 201 | LogP↓+HBA↑ | 0.94 |
| 202 | LogP↑+HBA↑ | 0.93 |
| 203 | LogP↓+HBD↑ | 0.95 |
| 204 | LogP↑+HBD↑ | 0.96 |
| 205 | LogP↓+TPSA↓ | 0.92 |
| 206 | LogP↓+TPSA↑ | 0.94 |
| **Overall diversity** | | **0.88** |

**Scaffold Change Analysis.** Table 14 reports the frequency of scaffold-preserving versus scaffold-hopping edits stratified by action type, computed on 50,000 stratified random samples (95% confidence interval, ±1% error margin) via Bemis-Murcko scaffold decomposition. The results demonstrate that the coarse-grained action taxonomy (Delete, Insert, Replace) produces complex molecular transformations beyond simple functional group substitutions. Delete operations yield the highest scaffold hop rate (85.86%), with 60.97% involving ring modifications. Insert operations achieve 92.06% scaffold hops and introduce new rings in 74.22% of cases. Replace operations show 73.04% scaffold changes with diverse ring system modifications. These findings indicate that MEGA's edit operations capture substantial structural diversity, including scaffold hopping and ring system transformations that are critical for lead optimization in drug discovery.

Table 14: Scaffold changes by action type computed via Bemis-Murcko decomposition on 50K stratified samples.

| Action | Valid Pairs | Same Scaffold | Scaffold Hop | Ring Add | Ring Del | Ring Mod |
|---|---|---|---|---|---|---|
| Delete | 1,591 | 14.14% | 85.86% | 0.13% | 38.91% | 60.97% |
| Insert | 21,876 | 7.94% | 92.06% | 74.22% | 0.00% | 25.78% |
| Replace | 26,533 | 26.96% | 73.04% | 30.78% | 3.49% | 65.73% |

Mean shifts, Table 15, align with the instructions for every task. Examples: $LogP\downarrow$ (101) moves the mean by $-0.975$ (winners vs. parents) and separates winners from losers by $-1.577$; $LogP\uparrow$ (102) shifts by $+0.965$ with a winner–loser gap of $+1.133$; $QED\downarrow$ (104) shifts by $-0.217$; $TPSA\uparrow$ (106) exhibits a large increase of $+31.611$; $HBA\uparrow$ (107) and $HBD\uparrow$ (108) increase by $+2.749$ and $+2.316$, respectively. The consistent sign and sizable winner–loser separations (last column) provide evidence of strong task-wise consistency on MEGA.

| Task | Property | Obj. | Parent $\bar{x}$ | Winner $\bar{x}$ | $\Delta$ W–P | Loser $\bar{x}$ | $\Delta$ W–L |
|---|---|---|---|---|---|---|---|
| 101 | LogP | $\downarrow$ | 2.475 | 1.501 | $-0.975$ | 3.078 | $-1.577$ |
| 102 | LogP | $\uparrow$ | 2.475 | 3.440 | $+0.965$ | 2.307 | $+1.133$ |
| 103 | QED | $\uparrow$ | 0.733 | 0.797 | $+0.064$ | 0.614 | $+0.183$ |
| 104 | QED | $\downarrow$ | 0.733 | 0.516 | $-0.217$ | 0.727 | $-0.211$ |
| 105 | TPSA | $\downarrow$ | 64.918 | 49.669 | $-15.249$ | 77.022 | $-27.353$ |
| 106 | TPSA | $\uparrow$ | 64.918 | 96.530 | $+31.611$ | 61.857 | $+34.673$ |
| 107 | HBA | $\uparrow$ | 3.990 | 6.739 | $+2.749$ | 4.224 | $+2.515$ |
| 108 | HBD | $\uparrow$ | 1.237 | 3.553 | $+2.316$ | 1.248 | $+2.305$ |

Table 15: MEGA: mean target-property values and deltas. $\Delta_{\text{W–P}} = \bar{x}_{\text{W}} - \bar{x}_{\text{P}}$ (winners minus parents) and $\Delta_{\text{W–L}} = \bar{x}_{\text{W}} - \bar{x}_{\text{L}}$ (winners minus losers). "Winners" and "losers" correspond to successful and unsuccessful edits, on strict threshold respectively. Signs follow the task objective (increase/decrease).

**Target Property Shifts.** Figure 5 visualizes the single-objective shifts via kernel density estimates of the target property for parent (orange) and edited child (blue) molecules. Across all eight tasks, the child distribution moves in the instructed direction (reduce/increase or count increase), demonstrating strong task-wise consistency in MEGA.

**Property Trade-off Analysis.** Table 16 reports property trade-offs observed in MEGA's successful edits across additional molecular descriptors. While target properties move in the desired directions by construction, optimization often incurs trade-offs in orthogonal properties. Synthetic Accessibility (SA) increases across all tasks (+0.22 to +0.63), indicating that property-improving edits tend to add structural complexity. Molecular Weight (MW) increases substantially for most tasks (+9 to +80 Da), with the exception of QED↑ optimization (+10 Da). Drug-likeness (QED) decreases for most tasks except QED↑ itself, reflecting inherent tensions between individual property improvements and overall drug-like character. Scaffold retention varies from 10.1% to 25.8%, confirming that many successful edits involve scaffold hopping. These trade-offs provide researchers with stratified dataset subsets for targeted training objectives.

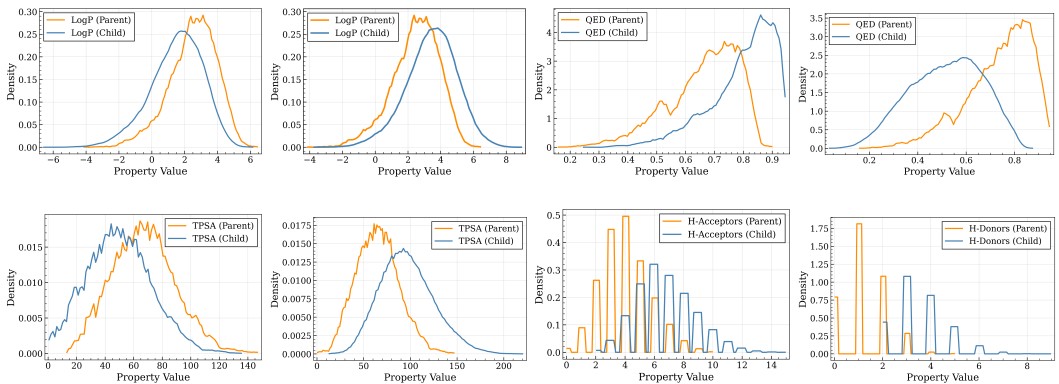

Figure 5: Molecular property distributions between parent and child molecules for MEGA.

Table 16: Property trade-offs in MEGA successful edits. $\Delta$ values show mean change from parent to child molecule; Scaffold % indicates Bemis-Murcko scaffold retention rate.

| Target | SA $\Delta$ | MW $\Delta$ | QED $\Delta$ | Scaffold % |
|---|---|---|---|---|
| MolLogP↑ | +0.52 | +55 | −0.19 | 25.0% |
| MolLogP↓ | +0.42 | +76 | −0.14 | 21.4% |
| QED↑ | +0.28 | +10 | +0.14 | 10.1% |
| QED↓ | +0.42 | +77 | −0.22 | 22.8% |
| TPSA↑ | +0.22 | +9 | −0.08 | 16.4% |
| TPSA↓ | +0.42 | +72 | −0.18 | 22.9% |
| HBA↑ | +0.50 | +80 | −0.19 | 16.4% |
| HBD↑ | +0.63 | +76 | −0.31 | 25.8% |

**Chemical Distance to Other Datasets.** For comparison to prior datasets, we report the Fréchet ChemNet Distance (FCD; lower is closer) (Preuer et al., 2018). As shown in Table 17, the distance between MolEdit and MolOpt roughly 4x lower compared to MEGA. This indicates that MEGA occupies a distinct region of the chemical space, while the incumbent datasets exhibit notable overlap, thus, expanding the resources available in the existing literature.

Table 17: Fréchet distance between datasets computed in Morgan-fingerprint space (lower is closer).

| Dataset | MEGA | MolEdit | MolOpt |
|---|---|---|---|
| MEGA | 0.000 | 2.790 | 2.738 |
| MolEdit | 2.790 | 0.000 | 0.696 |
| MolOpt | 2.738 | 0.696 | 0.000 |

## A.4 PROMPT TEMPLATES

___ **Prompt 101: Reduce** $\log P$ ___

**User:** Can you make molecule `SMILES` more soluble in water? The output molecule should be similar to the input molecule.
**Output:** One valid SMILES.

___ **Prompt 102: Increase** $\log P$ ___

**User:** Can you make molecule `SMILES` less soluble in water? The output molecule should be similar to the input molecule.
**Output:** One valid SMILES.

___ **Prompt 103: Increase** QED ___

**User:** Can you make molecule `SMILES` more like a drug? The output molecule should be similar to the input molecule.
**Output:** One valid SMILES.

___ **Prompt 104: Reduce** QED ___

**User:** Can you make molecule `SMILES` less like a drug? The output molecule should be similar to the input molecule.
**Output:** One valid SMILES.

___ **Prompt 105: Decrease** TPSA ___

**User:** Can you make molecule `SMILES` higher permeability? The output molecule should be similar to the input molecule.
**Output:** One valid SMILES.

___ **Prompt 106: Increase** TPSA ___

**User:** Can you make molecule `SMILES` lower permeability? The output molecule should be similar to the input molecule.
**Output:** One valid SMILES.

___ **Prompt 107: Increase** HBA ___

**User:** Can you make molecule `SMILES` with more hydrogen bond acceptors? The output molecule should be similar to the input molecule.
**Output:** One valid SMILES.

___ **Prompt 108: Increase** HBD ___

**User:** Can you make molecule `SMILES` with more hydrogen bond donors? The output molecule should be similar to the input molecule.
**Output:** One valid SMILES.

---
### Prompt 201: Reduce $\log P$ & Increase HBA
---

**User:** Can you make molecule `SMILES` more soluble in water and more hydrogen bond acceptors? The output molecule should be similar to the input molecule.

**Output:** One valid SMILES.

---
### Prompt 202: Increase $\log P$ & Increase HBA
---

**User:** Can you make molecule `SMILES` less soluble in water and more hydrogen bond acceptors? The output molecule should be similar to the input molecule.

**Output:** One valid SMILES.

---
### Prompt 203: Reduce $\log P$ & Increase HBD
---

**User:** Can you make molecule `SMILES` more soluble in water and more hydrogen bond donors? The output molecule should be similar to the input molecule.

**Output:** One valid SMILES.

---
### Prompt 204: Increase $\log P$ & Increase HBD
---

**User:** Can you make molecule `SMILES` less soluble in water and more hydrogen bond donors? The output molecule should be similar to the input molecule.

**Output:** One valid SMILES.

---
### Prompt 205: Reduce $\log P$ & Decrease TPSA
---

**User:** Can you make molecule `SMILES` more soluble in water and higher permeability? The output molecule should be similar to the input molecule.

**Output:** One valid SMILES.

---
### Prompt 206: Reduce $\log P$ & Increase TPSA
---

**User:** Can you make molecule `SMILES` more soluble in water and lower permeability? The output molecule should be similar to the input molecule.

**Output:** One valid SMILES.

# B GENERALIZATION TO UNSEEN PROPERTIES

This appendix presents additional evaluation results demonstrating that models trained on MEGA learn generalizable molecular editing skills that transfer beyond the RDKit-calculable properties used during training. We evaluate MEGA-GRPO (Llama-3-8B post-trained with MEGA using Group Relative Policy Optimization with similarity-aware rewards) on two categories of unseen objectives: (1) biological endpoints requiring docking simulations or QSAR models, (2) held-out multi-objective task combinations excluded from training.

**Evaluation Protocol.** For biological endpoints and held-out tasks, we follow the ChatDrug evaluation protocol (Liu et al., 2023b), reporting three metrics: **Success Rate (SR%)** measures the proportion of valid SMILES generations; **Hit Rate Loose (HR_L%)** and **Hit Rate Strict (HR_S%)** measure the proportion of generations meeting property improvement thresholds under loose and strict criteria, respectively. All evaluations use single-shot generation (one attempt per input molecule) without iterative refinement, representing the most challenging evaluation setting.

## B.1 ZERO-SHOT PERFORMANCE ON BIOLOGICAL ENDPOINTS

Table 18 reports zero-shot performance on three biological endpoints that were never seen during training and cannot be computed with RDKit: Dopamine D2 Receptor (DRD2), c-Jun N-terminal Kinase 3 (JNK3), and Glycogen Synthase Kinase 3$\beta$ (GSK3$\beta$). These endpoints require docking simulations or QSAR models, representing a fundamentally different evaluation paradigm from the RDKit-based properties in MEGA. MEGA-GRPO achieves 83.0% mean success rate and 46.3% mean hit rate (loose threshold), representing +76.0pp and +43.1pp improvements over the base Llama-3-8B model, respectively. Under strict thresholds, MEGA-GRPO achieves 43.5% mean hit rate versus 1.5% for the base model (+42.0pp improvement). These substantial gains demonstrate that training on MEGA enables models to learn core molecular editing patterns that generalize to pharmacologically relevant endpoints beyond the training distribution.

Table 18: Zero-shot performance on biological endpoints. SR = Success Rate, HR_L = Hit Rate (loose threshold), HR_S = Hit Rate (strict threshold).

| Model | Task | SR% | HR_L% | HR_S% |
|---|---|---|---|---|
| Llama-3-8B (MEGA) | DRD2↑ | 84.5 | 58.5 | 54.5 |
| Llama-3-8B (BASE) | | 9.5 | 4.5 | 1.0 |
| Llama-3-8B (MEGA) | JNK3↑ | 81.0 | 29.5 | 27.5 |
| Llama-3-8B (BASE) | | 5.5 | 1.5 | 1.5 |
| Llama-3-8B (MEGA) | GSK3B↑ | 83.5 | 51.0 | 48.5 |
| Llama-3-8B (BASE) | | 6.0 | 3.5 | 2.0 |
| **Mean (MEGA)** | | **83.0** | **46.3** | **43.5** |
| **Mean (BASE)** | | **7.0** | **3.2** | **1.5** |

## B.2 HELD-OUT MULTI-OBJECTIVE TASKS

Table 19 reports performance on multi-objective optimization tasks entirely excluded from the MEGA training dataset. These task combinations (LogP↑+TPSA↓, QED↑+TPSA↑, HBA↑+TPSA↑) were deliberately held out to evaluate compositional generalization—the ability to combine learned property optimization strategies in novel ways. MEGA-GRPO achieves 90.3% mean success rate and 54.7% mean hit rate (loose threshold), compared to 5.5% and 2.2% for the base model, respectively. Under strict thresholds, MEGA-GRPO achieves 41.5% mean hit rate versus 0.2% for the base model, representing a +41.3pp improvement. These results demonstrate that MEGA-trained models can compositionally combine optimization objectives not explicitly seen during training, indicating that the model has learned decomposable molecular editing strategies rather than memorizing task-specific patterns.

Table 19: Performance on held-out multi-objective tasks excluded from training. Task combinations test compositional generalization of learned editing skills.

| Model | Task | SR% | HR_L% | HR_S% |
|---|---|---|---|---|
| Llama-3-8B (MEGA) | LogP↑+TPSA↓ | 90.5 | 54.0 | 40.5 |
| Llama-3-8B (BASE) | | 2.0 | 0.0 | 0.0 |
| Llama-3-8B (MEGA) | QED↑+TPSA↑ | 86.5 | 18.0 | 2.0 |
| Llama-3-8B (BASE) | | 6.0 | 0.0 | 0.0 |
| Llama-3-8B (MEGA) | HBA↑+TPSA↑ | 94.0 | 92.0 | 82.0 |
| Llama-3-8B (BASE) | | 8.5 | 6.5 | 0.5 |
| **Mean (MEGA)** | | **90.3** | **54.7** | **41.5** |
| **Mean (BASE)** | | **5.5** | **2.2** | **0.2** |

## C  EXTRA COMPARISONS

### C.1  CHEMCOTBENCH EXTERNAL BENCHMARK

To further assess the utility of the MEGA dataset and the robustness of our findings beyond dataset-aligned evaluation tasks, we evaluated MEGA-GRPO on ChemCoTBench (Li et al., 2025), an independent benchmark for molecular optimization that includes 23 strong commercial, open-source, and domain-specialized models. Table 20 reports performance on three molecular properties (LogP, Solubility, QED) using the benchmark's native evaluation metrics: property change ($\Delta$) and success rate (SR%). The benchmark evaluates models in two categories: those with extended reasoning capabilities (8 models including DeepSeek-R1, Gemini-2.5-pro-think, o3-mini) and those without (15 models including GPT-4o, Gemini-2.0-flash, DeepSeek-V3). MEGA-GRPO achieves the best overall performance across all 24 models with an average $\Delta$ of 0.90 and 93.3% mean success rate. Notably, MEGA-GRPO achieves these results without extended reasoning capabilities, using only single-shot generation, yet outperforms all reasoning-enabled systems including Gemini-2.5-pro-think ($\Delta = 0.61$, SR = 85.7%), DeepSeek-R1 ($\Delta = 0.63$, SR = 81.0%), and o3-mini ($\Delta = 0.54$, SR = 79.7%). The model demonstrates particularly strong performance on LogP optimization ($\Delta = 1.46$, SR = 100%) while maintaining competitive results on Solubility ($\Delta = 1.10$, SR = 99%) and QED ($\Delta = 0.15$, SR = 81%). These results on an external benchmark with diverse baseline comparisons, including state-of-the-art reasoning models and domain-specialized systems, confirm that MEGA enables training of models with leading molecular optimization capabilities.

### C.2  EXTRA COMPARISONS WITH OTHER DATASETS

We extend the results in Table 2, we conducted an pair-wise comparison between MEGA and each external dataset on their overlapping task sets. Specifically, MEGA shares five tasks with MolEdit-Instruct and six with MolOpt-Instructions.

In the first experience, we trained models exclusively on the five tasks shared between MEGA and MolEdit-Instruct, namely tasks 103, 104, 107, 108, and 201 (Table 21). This setting corresponds to 678K training examples from MolEdit-Instruct and 183K examples from MEGA restricted to these five tasks. In the second, we trained models on the six tasks shared between MEGA and MolOpt-Instructions, namely tasks 101, 102, 103, 104, 107, and 108 (Table 22), which amounts to 301K training examples from MolOpt-Instructions and 229K examples from MEGA. All training hyperparameters and conditions described in Appendix B were kept identical to ensure a fair and controlled comparison.

In these head-to-head evaluations, we found that models trained on the MEGA data partitions, in average, outperform those trained on the corresponding data from MolEdit-Instruct and MolOpt-Instructions. This finding further validates the quality and effectiveness of our dataset, demonstrating that its superior performance is not limited to a small task intersection, but holds true in expanded comparisons.

Table 20: ChemCoTBench external validation results comparing MEGA-GRPO against 23 commercial, open-source, and domain-specialized models. Models are categorized by reasoning capability. $\Delta$ indicates mean property change; SR% indicates success rate.

| Models | LogP | | Solubility | | QED | | Average | |
|---|---|---|---|---|---|---|---|---|
| | $\Delta$ | SR% | $\Delta$ | SR% | $\Delta$ | SR% | $\Delta$ | SR% |
| *W/ Thinking* | | | | | | | | |
| DeepSeek-R1 | 0.36 | 74 | 1.48 | 97 | 0.05 | 72 | 0.63 | 81.0 |
| Gemini-2.5-pro-think | -0.28 | 81 | **1.91** | 92 | **0.21** | 84 | 0.61 | 85.7 |
| o3-mini@20250103 | 0.29 | 68 | 1.15 | 85 | 0.17 | **86** | 0.54 | 79.7 |
| o1-mini@20240912 | -0.42 | 52 | 1.78 | 95 | 0.07 | 70 | 0.48 | 72.3 |
| Claude3.7-sonnet-think | 0.41 | 81 | 0.59 | 77 | 0.09 | 73 | 0.36 | 77.0 |
| Qwen3-235B-A22B-think | -0.01 | 41 | 0.27 | 42 | 0.01 | 24 | 0.09 | 35.7 |
| Qwen3-32B-think | 0.00 | 2 | 0.11 | 23 | 0.02 | 14 | 0.04 | 13.0 |
| Llama-Nemo-49B-think | -0.64 | 24 | 0.20 | 24 | -0.16 | 41 | -0.20 | 29.7 |
| *W/o Thinking* | | | | | | | | |
| **MEGA-GRPO (Ours)** | **1.46** | **100** | 1.10 | **99** | 0.15 | 81 | **0.90** | **93.3** |
| GPT-4o@20241120 | -0.20 | 42 | 0.82 | 80 | 0.05 | 70 | 0.22 | 64.0 |
| Gemini-2.0-flash | 0.35 | 75 | 0.19 | 54 | 0.10 | 79 | 0.21 | 69.3 |
| DeepSeek-V3 | 0.08 | 34 | 0.47 | 93 | 0.08 | 46 | 0.21 | 57.7 |
| Qwen3-235B-A22B | 0.02 | 41 | 0.51 | 45 | 0.01 | 26 | 0.18 | 37.3 |
| Llama-3.3-70B-Instruct | -0.16 | 42 | 0.61 | 80 | 0.07 | 61 | 0.17 | 61.0 |
| Qwen2.5-32B-Instruct | 0.03 | 47 | 0.42 | 66 | -0.01 | 54 | 0.15 | 55.7 |
| Gemma-2-27b-it | -0.03 | 34 | 0.34 | 66 | 0.05 | 56 | 0.12 | 52.0 |
| Phi-4-14B | -0.10 | 45 | 0.28 | 54 | 0.11 | 74 | 0.10 | 57.7 |
| BioMistral-7B | 0.01 | 1 | 0.24 | 6 | 0.00 | 0 | 0.08 | 2.3 |
| Qwen2.5-72B-Instruct | -0.12 | 42 | 0.28 | 60 | 0.03 | 57 | 0.06 | 53.0 |
| Llama-Nemo-Super-49B | -0.14 | 27 | 0.31 | 41 | 0.02 | 50 | 0.06 | 39.3 |
| Qwen3-32B | -0.03 | 2 | 0.17 | 23 | 0.02 | 14 | 0.05 | 13.0 |
| BioMedGPT-7B | -0.36 | 17 | 0.25 | 63 | -0.29 | 7 | -0.13 | 29.0 |
| OLMo2-32B-Instruct | -2.03 | 22 | 1.03 | 46 | -0.13 | 40 | -0.38 | 36.0 |

Table 21: Performance comparison: MEGA vs MolEdit Instruct

| Task | Threshold | MolEdit-Instruct | MEGA |
|---|---|---|---|
| 103 | 0.0 | 27.19 ± 0.84 | **61.05 ± 2.88** |
| | 0.1 | 14.37 ± 0.95 | **24.36 ± 1.73** |
| 104 | 0.0 | **99.28 ± 0.52** | 95.84 ± 0.89 |
| | 0.1 | **97.94 ± 0.55** | 80.95 ± 3.41 |
| 107 | 0.0 | 95.72 ± 0.61 | **98.02 ± 0.90** |
| | 1.0 | 43.05 ± 1.64 | **94.58 ± 0.76** |
| 108 | 0.0 | 98.10 ± 0.71 | **99.80 ± 0.25** |
| | 1.0 | 66.53 ± 2.05 | **97.25 ± 0.60** |
| 201 | 0.0 | 87.14 ± 1.99 | **96.18 ± 1.03** |
| | 0.5 | 81.66 ± 1.72 | **87.86 ± 1.58** |
| **Average** | | 71.10 | **83.59** |

Table 22: Performance comparison: MEGA vs MolOpt-Instructions

| Task | Threshold | MolOpt-Instruction | MEGA |
|---|---|---|---|
| 101 | 0.0 | 96.71 ± 0.70 | **98.04 ± 0.51** |
| | 0.5 | **96.41 ± 0.58** | 92.47 ± 0.79 |
| 102 | 0.0 | 88.41 ± 1.85 | **97.41 ± 0.74** |
| | 0.5 | 88.41 ± 1.85 | **92.53 ± 2.25** |
| 103 | 0.0 | 16.82 ± 1.57 | **59.71 ± 1.29** |
| | 0.1 | 8.68 ± 1.16 | **26.72 ± 2.31** |
| 104 | 0.0 | **97.92 ± 1.40** | 97.42 ± 0.32 |
| | 0.1 | **93.68 ± 1.88** | 84.54 ± 2.15 |
| 107 | 0.0 | 92.33 ± 2.42 | **98.35 ± 0.50** |
| | 1.0 | 33.41 ± 3.03 | **93.36 ± 0.57** |
| 108 | 0.0 | 94.76 ± 0.91 | **100.00 ± 0.00** |
| | 1.0 | 56.10 ± 1.74 | **98.56 ± 0.89** |
| **Average** | | 71.97 | **86.59** |

## D  MEGA MODELS POST-TRAINING DETAILS

### D.1  SUPERVISED FINE-TUNING (SFT) DETAILS

For all our fine-tuning experiments, we utilize a memory-efficient, 4-bit quantized LLaMA 3.1 8B Instruct model as the backbone. Our datasets are consistently formatted as prompt-completion pairs, where the prompts are detailed in the main text and the corresponding completions are the child SMILES.

To ensure a fair comparison across benchmarks, we trained three models, as detailed in Table 2, each on a different dataset that has been filtered to contain comparable tasks. For the MEGA dataset, we retain tasks 101, 102, 103, 104, 107, and 108, resulting in 229K prompt-completion pairs. For MolEdit-Instruct, we use tasks 103, 104, 107, and 108 (as tasks 101 and 102 are not available), yielding 650K prompt-completion pairs. For MolOpt-Instructions, we include tasks 101, 102, 103, 104, 107, and 108, producing 301K prompt-completion pairs.

All models are trained using Low-Rank Adaptation (LoRA) with a rank of $r$=32 and $\alpha$=16, targeting all attention projection matrices and feed-forward layers. We use a training batch size of 16 with a gradient accumulation of 2 steps, resulting in an effective batch size of 32. Optimization is performed with an 8-bit quantized AdamW optimizer for memory efficiency. The learning rate is set to $1e - 4$ with a cosine annealing scheduler and a linear warm-up period of 100 steps. For regularization, a weight decay of 0.01 is applied. All models are trained with a maximum sequence length of 512 tokens, using mixed-precision training (bfloat16) when supported. All trainings are conducted on a single A100 (40GB) GPU for approximately 23 hours.

**SFT Evaluation.**  We perform a sanity check to ensure that test SMILES are not present in any of the training sets using canonical SMILES notation to prevent any data leakage. Importantly, to ensure the fairest possible evaluation, we evaluate each model using the prompt templates specific to their respective training datasets. This means models trained on MolEdit-Instruct data are evaluated with MolEdit-Instruct prompt templates, while models trained on MEGA use MEGA templates, and models trained on MolOpt-Instructions use MolOpt-Instructions templates, eliminating any potential bias from prompt format differences.

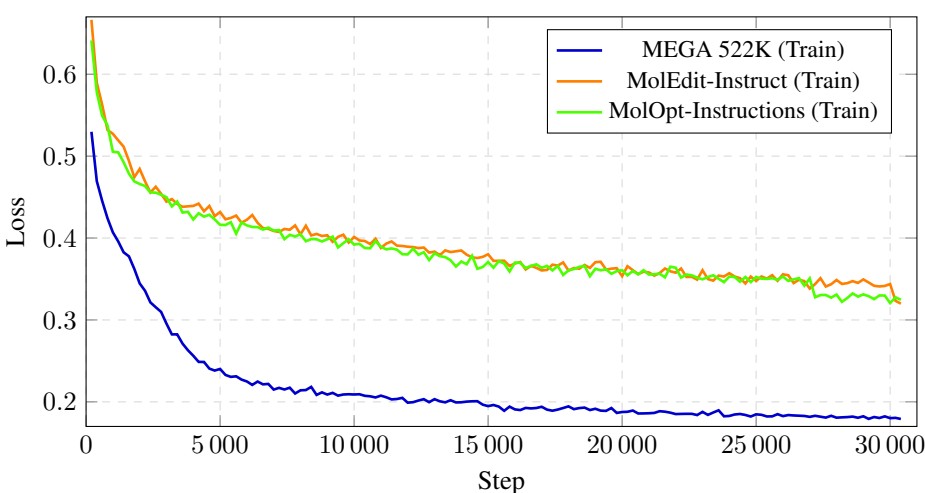

Figure 6: Training loss curves for three SFT models on MEGA, MolEdit-Instruct, and MolOpt-Instructions. MEGA achieves the lowest final loss (∼0.18), followed by MolOpt-Instructions and MolEdit-Instruct respectively.

As shown in Figure 6 the model trained on MEGA exhibits significantly faster convergence and substantially lower final loss values. The better training dynamics observed with MEGA indicates that our dataset leads to more sample-efficient learning, achieving better optimization faster.

In addition, for Table 3, we report hit ratio results comparing MEGA GRPO against Gemini 2.5 Pro (June 17, 2025 official API release) and DrugAssist on the 500 test SMILES provided by DrugAssist. This evaluation is performed over a single run, and we carefully verify that none of these 500 SMILES are included in our training set to avoid any possibility of data contamination.

### D.2  TANIMOTO-GRPO DETAILS

For locality-aware GRPO training, we ensured strict consistency between supervised fine-tuning (SFT) and post-training data. For example, the MEGA GRPO (14K) model used the same 14K SMILES for both SFT and GRPO. Similarly, the results in Table 4 were obtained from a policy model first fine-tuned on the full 522K prompt–completion pairs of MEGA, with the same data reused during GRPO. In this phase, we sampled $G = 12$ generations per prompt and computed rewards for each candidate molecule.

All Tanimoto-GRPO training was conducted on a single A100 GPU, with convergence achieved in approximately 10 hours at around 3,000 steps. We used an 8-bit quantized AdamW optimizer with a learning rate of $\alpha = 5 \times 10^{-6}$, $\beta_1 = 0.9$, $\beta_2 = 0.999$, a weight decay of 0.01, and gradient norm clipping at 0.5. The learning rate followed a cosine annealing schedule with a 10% linear warmup. To ensure memory efficiency, the model incorporated 4-bit quantization and LoRA adaptation with a rank of $r = 32$. We used an effective batch size of 8 (4 samples per device with 2 gradient accumulation steps) and maximum sequence lengths of 256 and 128 for prompts and completions, respectively. All computations were performed using bfloat16 mixed precision.

**Tanimoto-GRPO Algorithm.** For each molecular editing prompt $(x_{\mathrm{in}}, x_t)$, our Tanimoto-GRPO algorithm operates as follows:

1. Sample a group of candidate molecules:

$$\{y_1, y_2, ..., y_G\} \sim \pi_\theta(\cdot|x_{\mathrm{in}}, x_t) \tag{1}$$

where $G$ is the number of generations by our policy model

2. Compute composite rewards for all candidates:

$$r_i = R_{\mathrm{validity}}(y_i) \times R_{\mathrm{property}}(y_i, x_{\mathrm{in}}, x_t) \times R_{\mathrm{similarity}}(y_i, x_{\mathrm{in}}) \quad \text{for } i = 1, ..., G \tag{2}$$

where the reward is decomposed into three components detailed below.

3. Calculate group-relative advantages:

$$\hat{A}_i = \frac{r_i - \bar{r}}{\sigma_r + \epsilon} \tag{3}$$

where $\bar{r} = \frac{1}{G} \sum_{j=1}^{G} r_j$ and $\sigma_r = \sqrt{\frac{1}{G} \sum_{j=1}^{G} (r_j - \bar{r})^2}$ are the mean and standard deviation of rewards within the group, and $\epsilon = 10^{-8}$ for numerical stability.

4. Update the policy using the GRPO objective:

$$\mathcal{L}_{\mathrm{GRPO}}(\theta) = -\frac{1}{\sum_{i=1}^{G} |y_i|} \sum_{i=1}^{G} \sum_{t=1}^{|y_i|} \left[ \min\left(\rho_{i,t} \hat{A}_i, \mathrm{clip}(\rho_{i,t}, 1 - \varepsilon, 1 + \varepsilon) \hat{A}_i \right) - \beta D_{\mathrm{KL}}[\pi_\theta \| \pi_{\mathrm{ref}}] \right] \tag{4}$$

where:

- $\rho_{i,t} = \frac{\pi_\theta(y_{i,t}|x_{\mathrm{in}}, x_t, y_{i,<t})}{\pi_{\theta_{\mathrm{old}}}(y_{i,t}|x_{\mathrm{in}}, x_t, y_{i,<t})}$ is the probability ratio
- $\varepsilon = 0.2$ is the clipping parameter
- $\beta = 0.0$ by default
- If $\beta > 0$, the KL divergence is estimated as shown previously

**Reward Function Components.** The composite reward function consists of three multiplicative components, ensuring that all criteria must be satisfied for a candidate molecule to receive a positive reward. The validity reward $R_{\mathrm{validity}}(y)$ ensures generated molecules are chemically valid:

$$R_{\text{validity}}(y) = \begin{cases} 0 & \text{if } y \text{ contains fragments ("."}) \\ 0 & \text{if } y = x_{\text{in}} \text{ (no modification)} \\ 0 & \text{if } y \text{ fails RDKit sanitization} \\ 1 & \text{otherwise} \end{cases} \tag{5}$$

This component filters out invalid SMILES strings, fragmented molecules, and trivial identity transformations. The property reward $R_{\text{property}}(y, x_{\text{in}}, x_t)$ evaluates whether the generated molecule achieves the target property improvement specified by task $x_t$. For each task, we define loose and strict thresholds:

$$R_{\text{property}}(y, x_{\text{in}}, x_t) = \begin{cases} 1.0 & \text{if } \Delta p \geq \tau_{\text{strict}} \\ 0.5 & \text{if } 0 < \Delta p < \tau_{\text{strict}} \\ 0.0 & \text{otherwise} \end{cases} \tag{6}$$

where $\Delta p$ is the property change and $\tau_{\text{strict}}$ is the strict threshold. Table 23 reports the strict thresholds used for each task.

Table 23: Strict thresholds for property rewards in Tanimoto-GRPO. Loose threshold is always 0 (any improvement in correct direction).

| Task | Objective | Strict Threshold ($\tau_{\text{strict}}$) |
|------|-----------|-------------------------------------------|
| 101 | LogP↓ | $\Delta$LogP $< -0.5$ |
| 102 | LogP↑ | $\Delta$LogP $> +0.5$ |
| 103 | QED↑ | $\Delta$QED $> +0.1$ |
| 104 | QED↓ | $\Delta$QED $< -0.1$ |
| 105 | TPSA↓ | $\Delta$TPSA $< -10$ |
| 106 | TPSA↑ | $\Delta$TPSA $> +10$ |
| 107 | HBA↑ | $\Delta$HBA $> +1$ |
| 108 | HBD↑ | $\Delta$HBD $> +1$ |
| 201 | LogP↓ + HBA↑ | $\Delta$LogP $< -0.5$ **and** $\Delta$HBA $> +1$ |
| 202 | LogP↑ + HBA↑ | $\Delta$LogP $> +0.5$ **and** $\Delta$HBA $> +1$ |
| 203 | LogP↓ + HBD↑ | $\Delta$LogP $< -0.5$ **and** $\Delta$HBD $> +1$ |
| 204 | LogP↑ + HBD↑ | $\Delta$LogP $> +0.5$ **and** $\Delta$HBD $> +1$ |
| 205 | LogP↓ + TPSA↓ | $\Delta$LogP $< -0.5$ **and** $\Delta$TPSA $< -10$ |
| 206 | LogP↓ + TPSA↑ | $\Delta$LogP $< -0.5$ **and** $\Delta$TPSA $> +10$ |

For multi-objective tasks (201–206), the strict reward of 1.0 requires meeting *both* thresholds simultaneously, while the loose reward of 0.5 requires improvement in *both* properties (any positive change in the correct direction). The similarity reward $R_{\text{similarity}}(y, x_{\text{in}})$ encourages local edits by rewarding structural similarity between input and output molecules:

$$R_{\text{similarity}}(y, x_{\text{in}}) = \begin{cases} 1.0 & \text{if } \text{Tanimoto}(y, x_{\text{in}}) > 0.65 \\ 0.5 & \text{if } 0.4 \leq \text{Tanimoto}(y, x_{\text{in}}) \leq 0.65 \\ 0.0 & \text{otherwise} \end{cases} \tag{7}$$

where Tanimoto similarity is computed using RDKit fingerprints:

$$\text{Tanimoto}(y, x_{\text{in}}) = \frac{|\text{FP}(y) \cap \text{FP}(x_{\text{in}})|}{|\text{FP}(y) \cup \text{FP}(x_{\text{in}})|} \tag{8}$$

This component balances exploration (allowing structural changes) with locality (maintaining similarity to the parent molecule), aligning with medicinal chemistry practice where lead optimization typically involves incremental modifications. The final reward combines all three components multiplicatively:

$$r_i = R_{\text{validity}}(y_i) \times R_{\text{property}}(y_i, x_{\text{in}}, x_t) \times R_{\text{similarity}}(y_i, x_{\text{in}}) \tag{9}$$

This multiplicative formulation ensures that:

- Invalid molecules receive zero reward regardless of property improvements
- Molecules that fail to improve target properties receive zero reward
- Molecules with very low similarity (Tanimoto $< 0.4$) receive zero reward
- Maximum reward (1.0) requires validity, strict property threshold achievement, and high similarity
- Intermediate rewards (0.5 or 0.25) are possible for partial success

This reward structure guides the model toward producing valid, locally modified molecules that achieve meaningful property improvements aligned with the specified optimization objective.

# E  POST-TRAINED MEGA MODEL BEHAVIOR.

## E.1  SIMILARITY CONTROL VIA TANIMOTO REWARD

Figure 7 shows the results of LLM postraining across varying dataset sizes sampled from MEGA using GRPO with and without incorporating a Tanimoto similarity component into the reward system. When trained without the Tanimoto reward on small datasets, the models achieve high hit ratios but tend to alter the scaffold substantially, yielding molecules with low similarity to their parent compounds. As the dataset size increases, however, the model implicitly recovers the similarity distribution observed in the SFT baseline, ultimately reaching the target similarity regime even without an explicit reward signal. In contrast, when the Tanimoto reward is included, the model attains this small-edit regime with as few as 1.4k training examples (roughly 100 per task type).

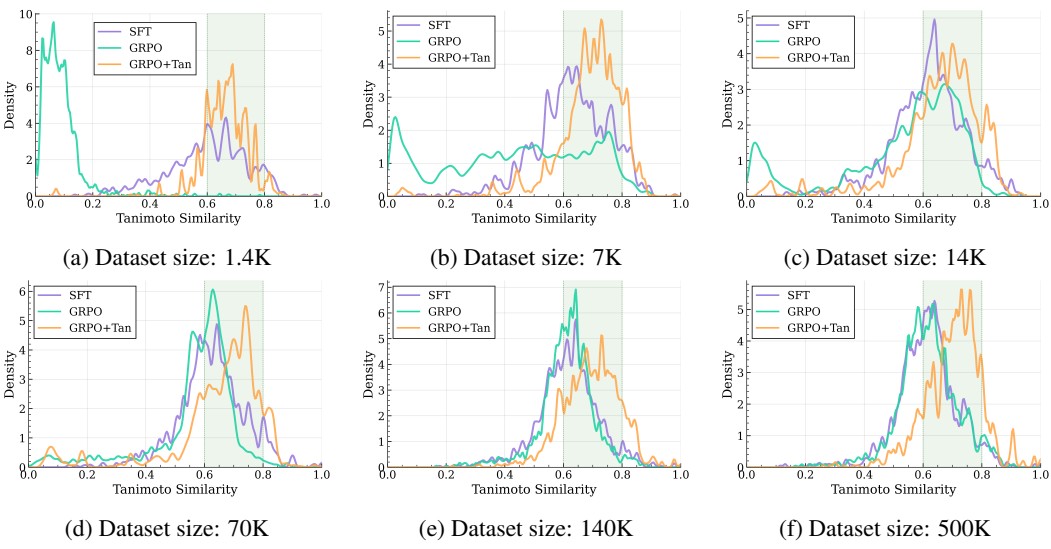

Figure 7: Tanimoto similarity distributions for different training data sizes. Each plot shows the distribution for SFT (purple), GRPO without Tanimoto reward (turquoise), and GRPO with Tanimoto reward (orange) models. The green shaded region (0.6–0.8) indicates the targeted tanimoto similarity range.

## E.2  CHEMICAL DIVERSITY OF MEGA MODEL OUTPUTS

**Transformation Diversity Analysis.**  To assess whether the Tanimoto-based similarity reward constrains model creativity, we analyzed the chemical diversity of MEGA-GRPO outputs on the

ChatDrug benchmark (200 SMILES, 14 optimization tasks), comprising 2,462 successful transformations. Table 24 reports the distribution of structural modification types. Despite the similarity constraint, 22.7% of transformations involve multi-site edits, 40.5% result in Bemis-Murcko scaffold hops, 22.6% modify ring systems, and the model generates 32 macrocycles from non-macrocyclic inputs. These results demonstrate that the Tanimoto reward does not restrict the model to trivial functional group substitutions, but rather guides it toward an appropriate balance between structural exploration and local optimization aligned with medicinal chemistry practice.

Table 24: Chemical diversity analysis of MEGA-GRPO outputs on the ChatDrug benchmark (2,462 successful transformations).

| Metric | Value |
| --- | --- |
| Total transformations | 2,462 |
| Single-site edits | 77.3% |
| Multi-site edits ($\geq$2 sites) | 22.7% |
| Bemis-Murcko Scaffold hops | 40.5% |
| Ring modifications | 22.6% |
| Macrocycles created (from non-macrocyclic inputs) | 32 |

### E.3 PROPERTY OPTIMIZATION STRATEGIES

**Threshold Exploitation Analysis.** To investigate whether GRPO-trained models exploit threshold boundaries rather than learning meaningful optimization strategies, we computed the distribution of property improvements relative to task-specific thresholds. Table 25 reports the percentage of successful transformations falling into three regions: near-threshold ($0$–$0.1\sigma$), moderate improvement ($0.1$–$1\sigma$), and large improvement ($>1\sigma$), where $\sigma$ represents the standard deviation of property changes in the training data. Only 8.75% of successful cases cluster around threshold boundaries, while 39% achieve large improvements exceeding $1\sigma$ above the required threshold. For unsuccessful attempts, 61.7% moved properties in the correct direction but failed to reach the threshold. Furthermore, 87.5% of tasks exhibit non-normal improvement distributions (Shapiro-Wilk test, $p < 0.05$), indicating the model employs diverse optimization strategies rather than converging to a single threshold-exploiting solution. These patterns demonstrate that MEGA-GRPO produces meaningful chemical transformations along property-relevant optimization trajectories.

Table 25: Distribution of property improvements relative to task thresholds for MEGA-GRPO outputs. $\sigma$ denotes standard deviation of property changes in training data.

| Category | Percentage |
| --- | --- |
| *Successful transformations:* | |
| Near threshold ($0$–$0.1\sigma$) | 8.75% |
| Moderate improvement ($0.1$–$1\sigma$) | 52.25% |
| Large improvement ($>1\sigma$) | 39.0% |
| *Failed transformations:* | |
| Moved in correct direction | 61.7% |
| Moved in wrong direction | 38.3% |
| Tasks with non-normal distributions | 87.5% |

**Optimization Trajectory Diversity.** The non-normal distributions and wide spread of improvements indicate that GRPO-trained models learn multiple optimization pathways for each task rather than memorizing single strategies. This diversity in optimization trajectories suggests robust generalization capabilities, as evidenced by the strong zero-shot transfer to biological endpoints and held-out task combinations reported in Appendix B.

# F    QUALITATIVE EXAMPLES

Table 26: Visualization of molecular editing with three actions: Replace, Insert, and Delete. The yellow regions indicate replaced substructures, the blue regions indicate inserted substructures, and the red regions indicate deleted substructures. Each example shows the transformation from the input molecule $\mathbf{x}_{in}$ to the output molecule $\mathbf{x}_{out}$.

| (a) 101 (strict) | (b) 106 (strict) |
|---|---|
| Input Molecule $\mathbf{x}_{in}$ → Output Molecule $\mathbf{x}_{out}$ | Input Molecule $\mathbf{x}_{in}$ → Output Molecule $\mathbf{x}_{out}$ |

LogP: 3.3398

$\rightarrow$ 2.2743  TPSA: 79.3700 $\rightarrow$ 103.1600

| (c) 102 (strict) | (d) 103 (loose) |
|---|---|
| Input Molecule $\mathbf{x}_{in}$ → Output Molecule $\mathbf{x}_{out}$ | Input Molecule $\mathbf{x}_{in}$ → Output Molecule $\mathbf{x}_{out}$ |

LogP: 1.6861

$\rightarrow$ 3.2998  QED: 0.8626 $\rightarrow$ 0.9025

| (e) 105 (strict) | (f) 107 (loose) |
|---|---|
| Input Molecule $\mathbf{x}_{in}$ → Output Molecule $\mathbf{x}_{out}$ | Input Molecule $\mathbf{x}_{in}$ → Output Molecule $\mathbf{x}_{out}$ |

TPSA:

89.3500 $\rightarrow$ 72.2800  H-Bond Acceptors: 2 $\rightarrow$ 3

| (g) 108 (strict) | (h) 205 (strict) |
|---|---|
| Input Molecule $\mathbf{x}_{in}$ → Output Molecule $\mathbf{x}_{out}$ | Input Molecule $\mathbf{x}_{in}$ → Output Molecule $\mathbf{x}_{out}$ |

H-Bond

Donors: 1 $\rightarrow$ 3  LogP: 3.0216 $\rightarrow$ 1.3313, TPSA: 44.81 $\rightarrow$ 32.18

| (i) 101 (strict) | (j) 102 (strict) |
|---|---|
| Input Molecule $\mathbf{x}_{in}$ → Output Molecule $\mathbf{x}_{out}$ | Input Molecule $\mathbf{x}_{in}$ → Output Molecule $\mathbf{x}_{out}$ |

LogP: 0.4971

→ -0.0816  LogP: 4.0895 → 4.6941

| (k) 104 (strict) | (l) 201 (strict) |
|---|---|
| Input Molecule $\mathbf{x}_{in}$ → Output Molecule $\mathbf{x}_{out}$ | Input Molecule $\mathbf{x}_{in}$ → Output Molecule $\mathbf{x}_{out}$ |

QED: 0.3421

→ 0.1626  LogP: 0.4971 → -0.1731, H-Acceptors: 9 → 11

| (m) 202 (strict) | (n) 204 (strict) |
|---|---|
| Input Molecule $\mathbf{x}_{in}$ → Output Molecule $\mathbf{x}_{out}$ | Input Molecule $\mathbf{x}_{in}$ → Output Molecule $\mathbf{x}_{out}$ |

LogP: 3.7027

→ 4.9789, H-Acceptors: 8 → 10  LogP: 5.7082 → 6.4651, H-Donors: 1 → 3

| (o) 206 (strict) | (p) 108 (strict) |
|---|---|
| Input Molecule $\mathbf{x}_{in}$ → Output Molecule $\mathbf{x}_{out}$ | Input Molecule $\mathbf{x}_{in}$ → Output Molecule $\mathbf{x}_{out}$ |

LogP: 4.0895

→ 3.3751, TPSA: 45.67 → 83.72  H-Bond Donors: 0 → 3

| (q) 102 (strict) | (r) 103 (strict) |
|---|---|
| Input Molecule $\mathbf{x}_{in}$ → Output Molecule $\mathbf{x}_{out}$ | Input Molecule $\mathbf{x}_{in}$ → Output Molecule $\mathbf{x}_{out}$ |

LogP: 3.0114

→ 3.5138  QED: 0.5656 → 0.8620

| (s) 105 (strict) | (t) 105 (strict) |
|---|---|
| Input Molecule $\mathbf{x}_{in}$ → Output Molecule $\mathbf{x}_{out}$ | Input Molecule $\mathbf{x}_{in}$ → Output Molecule $\mathbf{x}_{out}$ |

TPSA:

31.3500 → 18.4600  TPSA: 55.4000 → 38.3300

# G  DATASET LICENSE

We derived our work from the ZINC 250K dataset Akhmetshin et al. (2021), available here, which is distributed under the GNU General Public License v3 or later (GPL-3.0+). In accordance with this license, we release our derived dataset under the same terms, preserving the freedoms to use, share, and modify the data.

