# OpenReview forum: "MEGA: A Large-Scale Molecular Editing Dataset for Guided-Action Optimization"
_ICLR.cc/2026/Conference — Submitted to ICLR 2026_

### Official Review · Reviewer_gQsF · 2025-10-27

**Soundness:** 3
**Presentation:** 3
**Contribution:** 3
**Rating:** 6
**Confidence:** 4

**Summary:**

The authors release MEGA, a molecular editing dataset of 31.4M parent–child SMILES pairs (MEGA‑Large) with single action annotations (Insert/Delete/Replace) across 28 tasks; a 522k MEGA subset mirrors the action distribution for resource-constrained use. The construction uses common slicing rules to identify edit sites and validates property improvements deterministically via RDKit under MoleculeSTM-style thresholds. The dataset also provides 41M “near-miss” negatives to support contrastive and RL training. Using a fixed Llama‑3 8B with LoRA, the authors show SFT on MEGA improves hit ratios over other corpora, and GRPO post-training with a similarity-aware reward yields SOTA performance. The dataset is intended for open access.

**Strengths:**

* Scale & annotation: Largest-to-date positive edit corpus with Insert/Delete/Replace labels; supports per‑action supervision and diagnostics.
* Evidence of usefulness: Consistent SFT gains over other datasets on shared tasks, and strong GRPO results with similarity-aware rewards.
* Design for locality: Reward shaping explicitly balances property improvement and local edits, aligning with medicinal-chemistry practice.

**Weaknesses:**

* Proxy-based labels: All property “hits” are deterministic RDKit thresholds; consider releasing parallel subsets with more physically grounded or experimental endpoints where feasible.
* Scaffold drift: Construction “does not constrain scaffold preservation”; the dataset might favor edits that break similarity. Consider providing scaffold-retaining slices or tags to enable stratified training.
* Coverage analysis: Missing statistics on duplicate parents/children, class imbalance per action/task, and measures of chemical diversity across tasks.
* Licensing & reproducibility: Ensure licensing for derived data is explicit. Will scripts to regenerate thresholds and task splits be included?

**Questions:**

* Can you release per-action difficulty metrics (e.g., hit ratio vs. action, per-task) to encourage targeted benchmark ablations?
* How often do edits decrease other key properties (e.g., SA score) while optimizing the target? Consider multi-objective annotations for trade-off analysis.
* Will you publish a canonical evaluation server with fixed parents per task and verifiable calculators to avoid drift?

---

> ### Author Response · Authors · 2025-11-19
> **We thank Reviewer gQsF for their thorough evaluation and helpful feedback**
>
> ## General comments
>
> **Parallel subsets of physically grounded or experimental endpoints**: We thank the reviewer for this suggestion. As MEGA is a synthetic dataset, it is not possible to directly obtain experimental data from public databases. Nevertheless, in a future release we plan to annotate a small, representative subset of MEGA with ADMET properties via ADMET-AI.
>
> **Further MEGA statistics**: At the request of the reviewer we have added three new tables to [Appendix A](https://openreview.net/pdf?id=wzou4rm3Tt#page=14):
>
> - *Per-task action distribution:*  Action distributions vary by optimization objective, from balanced Insert/Replace (~50/50 for solubility tasks) to highly skewed patterns (Task 103: 77% Replace; Task 105: 0% Insert; Task 205: 97% Replace).
>
> - *Duplication statistics:* Parents show 99.2% reuse (ZINC250 original SMILES), while children show a mean of 30.2% duplication from convergent edits across tasks.
>
> - *Chemical diversity per task:* Scaffold diversity computed via Bemis-Murcko on 50k stratified samples (95% CI, ±1% margin) ranges from 0.91-0.98 per task, with overall diversity of 0.88, confirming substantial structural variation.
>
> **Licensing**: The licence details are provided in  [Appendix G](https://openreview.net/pdf?id=wzou4rm3Tt#page=31) of our manuscript.
>
> ## Q1. Can you release per-action difficulty metrics to encourage targeted benchmark ablations?
> In Table 1 we report the exploration efficiency as the average number of edits exceeding the task-wise strict thresholds per exploration episode, stratified by action type. These metrics are computed over **10.4M exploration episodes** (249,455 per task-action combination) spanning **11.3B oracle calls** to RDKit (NFE, Number of Function Evaluations). Table 1 shows the exploration yields computed as the number of successful edits for each task-action combination. Lower values indicates greater search effort, a natural proxy for task-action difficulty.
>
> **Table 1: Per-task exploration efficiency (Appendix A, page 16).**
>
> | Task | Objective | DELETE | INSERT | REPLACE |
> |------|-----------|-----|-----|-----|
> | 101  | LogP↓ | 0.61 (753K) | 4.93 (15.0M) | 4.94 (10.9M) |
> | 102  | LogP↑ | 0.59 (753K) | 4.93 (9.0M) | 4.94 (8.7M) |
> | 103  | QED↑ | 0.37 (753K) | 0.61 (1,798M) | 3.27 (2,013M) |
> | 104  | QED↓ | 0.43 (753K) | 4.93 (8.1M) | 4.94 (8.3M) |
> | 105  | TPSA↓ | 1.70 (751K) | 0.00 (1,923M) | 4.89 (759M) |
> | 106  | TPSA↑ | 0.00 (753K) | 4.93 (7.6M) | 4.94 (5.6M) |
> | 107  | HBA↑ | 0.00 (753K) | 4.93 (7.3M) | 4.94 (14.4M) |
> | 108  | HBD↑ | 0.00 (753K) | 4.93 (23.5M) | 4.94 (44.2M) |
> | 201  | LogP↓+HBA↑ | 0.00 (753K) | 4.93 (15.7M) | 4.94 (47.4M) |
> | 202  | LogP↑+HBA↑ | 0.00 (753K) | 4.93 (11.6M) | 4.92 (150.3M) |
> | 203  | LogP↓+HBD↑ | 0.00 (753K) | 4.93 (29.2M) | 4.94 (55.3M) |
> | 204  | LogP↑+HBD↑ | 0.00 (753K) | 4.90 (279.0M) | 4.73 (923.8M) |
> | 205  | LogP↓+TPSA↓ | 0.15 (753K) | 0.00 (1,923M) | 4.73 (1,214M) |
> | 206  | LogP↓+TPSA↑ | 0.00 (753K) | 4.93 (15.1M) | 4.94 (14.8M) |
> | **Overall** | **(all)** | **0.28 (10.5M)** | **3.92 (6.1B)** | **4.79 (5.3B)** |
>
>
>  ## Q2. Consider multi-objective annotations for trade-off analysis.
> At the reviewer's request, we have enriched MEGA with additional property annotations including **Synthetic Accessibility (SA)**, **Molecular Weight (MW)**, **drug-likeness (QED)** and the previously suggested **Murcko scaffold retention** annotations. Table 2 shows that while all targets move in the desired directions, overall, property trade-offs are common.
>
> | Target | SA_Δ   | MW_Δ  | QED_Δ  | Scaffold % |
> |-----------------|--------|-------|--------|------------|
> | MolLogP↑        | +0.52  | +55   | -0.19  | 25.0%      |
> | MolLogP↓        | +0.42  | +76   | -0.14  | 21.4%      |
> | QED↑            | +0.28  | +10   | +0.14  | 10.1%      |
> | QED↓            | +0.42  | +77   | -0.22  | 22.8%      |
> | TPSA↑           | +0.22  | +9    | -0.08  | 16.4%      |
> | TPSA↓           | +0.42  | +72   | -0.18  | 22.9%      |
> | HBA↑            | +0.50  | +80   | -0.19  | 16.4%      |
> | HBD↑            | +0.63  | +76   | -0.31  | 25.8%      |
>
> Using these annotations researchers now have access to sizeable partitions tailored for different purposes, including:
> - **Scaffold-preserving subset (5.7M pairs)**: Structure-preserving pairs that retain Murcko scaffold.
> - **Low-trade-off subset (3M pairs)** : Target property improves without substantial degradation of other key properties (SA_Δ < 0.3, MW_Δ < 50, QED_Δ > -0.1).
> - **Successful edits (31M pairs)**: Ideal for pre-training foundation models with broad molecular editing capabilities.
>
> Further details are available in [Appendix A.1, A.3](https://openreview.net/pdf?id=wzou4rm3Tt#page=14).
>
> ## Q3. Will you publish a canonical evaluation server?
> For flexibility of use, we release the dataset partitions and the MEGA-GRPO model weights on Hugging Face. We also release a GitHub repository with evaluation scripts and pinned versions for reproducibility.

---

### Official Review · Reviewer_wdBR · 2025-10-28

**Soundness:** 2
**Presentation:** 3
**Contribution:** 2
**Rating:** 2
**Confidence:** 4

**Summary:**

This paper introduces MEGA, a new family of large-scale datasets for molecular editing, comprising up to 31.4 million property-improving molecule pairs. A key feature of MEGA is that each pair is annotated with an explicit edit action: Replace, Insert, or Delete. The authors show that a model fine-tuned on MEGA (SFT) outperforms models trained on existing datasets. They additionally use GRPO with a similarity-aware reward for post-training, achieving new SOTA performance on several benchmarks.

**Strengths:**

1. The authors systematically validate the dataset's effectiveness through a well-designed two-stage process involving SFT and GRPO. The results convincingly demonstrate that models trained on MEGA achieve significant performance gains.
2. The paper innovatively reframes the complex task of molecular property optimization into "edit actions" that are more interpretable and easier for LLMs. The superior performance of models trained on MEGA compared to those trained on other datasets effectively validates the efficacy of this novel paradigm.
3. The authors release MEGA in open access to the community, which provides a solid foundation for reproducibility and future research.

**Weaknesses:**

1. While the MEGA dataset is a core contribution, the molecular properties it covers are narrowly focused on physicochemical attributes (e.g., LogP, TPSA, QED). These properties represent only a small fraction of the optimization objectives in drug discovery and can often be predicted with high accuracy using classical computational methods or rule-based models. The true value of AI in molecular optimization lies in addressing properties that are expensive to acquire experimentally, such as biological activity and ADMET profiles. To enhance the dataset's practical impact, it is strongly recommended that the authors consider extending it to include more critical endpoints like binding affinity and ADMET properties, which are abundantly available in public databases like ChEMBL.
2. Although the paper presents a comparison against DrugAssist and Gemini 2.5 Pro on the DrugAssist benchmark, it lacks a broader evaluation against other leading commercial and open-source LLMs and domain-specific models. Several public benchmarks for molecular optimization have already emerged (PMO, ChemCoTBench). Benchmarking the MEGA-trained models on these third-party platforms would be highly beneficial.

**Questions:**

1. The paper defines three coarse edit actions. While effective, this taxonomy might not capture more complex edits like scaffold hopping or ring system modifications. Have the authors considered these limitations?

---

> ### Author Response · Authors · 2025-11-18
> **We thank Reviewer wdBR for their thorough evaluation and address each concern below.**
>
> ## W1: “Consider extending MEGA endpoints using public databases like ChEMBL"
>
> We thank the reviewer for this suggestion. However, extending the dataset to include biological/ADMET endpoints is infeasible. MEGA contains ~72M algorithmically generated annotated edits around ZINC250. For the vast majority of these structures, no experimental measurements exist in ChEMBL, PubChem, or any other public source.
>
> More fundamentally, such extension is unnecessary for MEGA's objective: teaching LLMs strong and generalizable *molecular editing skills*. To demonstrate this capability, we evaluated MEGA-GRPO on three biological endpoints not present in RDKit: Dopamine D2 Receptor (DRD2), c-Jun N-terminal Kinase 3 (JNK3), and Glycogen Synthase Kinase 3β (GSK3β). A Llama 3-based model trained on MEGA achieved a **41% zero-shot hit rate improvement** compared to the baseline (see detailed results in  [Appendix B](https://openreview.net/pdf?id=wzou4rm3Tt#page=22)), demonstrating that the model learned generalizable skills that transfer to pharmacological endpoints not present in the training data.
>
> For researchers requiring explicit optimization on specific endpoints, the MEGA framework allows **natural extension without dataset modification**. During GRPO post-training, optimization relies solely on the *property oracle Δp* for rewards. As described in Section 4.2, the property-hit term $\mathbf{1}\left[\Delta p \ge \tau\right]$ depends on oracle output and is independent of SFT annotations. This means MEGA-trained models can be adapted for new endpoints as long as a property oracle exists, without needing extra annotated pairs.
>
> ## W2: Broader evaluation across leading models and benchmarks.
> In response to the reviewer’s helpful suggestion, we evaluated MEGA-GRPO on the relevant molecule optimization tasks of ChemCoTBench, which includes 23 strong commercial, open-source, and domain-specialized models. On this external benchmark, our 8B MEGA-GRPO achieves the **best overall performance**, and remains highly competitive in individual properties to much larger incumbents. We note that MEGA-GRPO favors similarity, which naturally bounds individual property changes.
>
> | Models                                            | Thinking | LogP Δ   | LogP SR% | Solubility Δ | Solubility SR% | QED Δ    | QED SR% | Average Δ | Average SR% |
> | ------------------------------------------------- | -------- | -------- | -------- | ------------ | -------------- | -------- | ------- | --------- | ----------- |
> | **MEGA-GRPO (Ours)**                              | ✗        | **1.46** | **100**  | 1.10         | **99**         | 0.15     | 81      | **0.90**  | **93.3**    |
> | Gemini-2.5-pro-think                              | ✓        | -0.28    | 81       | **1.91**     | 92             | **0.21** | 84      | 0.61      | 85.7        |
> | DeepSeek-R1                                       | ✓        | 0.36     | 74       | 1.48         | 97             | 0.05     | 72      | 0.63      | 81.0        |
> | o3-mini@20250103                                  | ✓        | 0.29     | 68       | 1.15         | 85             | 0.17     | **86**  | 0.54      | 79.7        |
> | Claude-3.7-Sonnet-Think                           | ✓        | 0.41     | 81       | 0.59         | 77             | 0.09     | 73      | 0.36      | 77.0        |
> | Gemini-2.0-flash (best non-thinking baseline) | ✗        | 0.35     | 75       | 0.19         | 79             | 0.15     | 63      | 0.21      | 69.3        |
>
> Regarding PMO, we respectfully note a methodological mismatch. PMO is designed for iterative evolutionary optimizers with hundreds-to-thousands of oracle calls, while MEGA-trained LLMs operate single-shot with one generation step. This makes direct comparison not methodologically aligned. We also note that our manuscript already evaluates MEGA-trained models across leading molecular optimization works (e.g., Liu et al., 2023a; Liu et al., 2023b; Zhuang et al., 2025; Ye et al., 2025; Comanici et al., 2025). See extended benchmarks in [Appendix C](https://openreview.net/pdf?id=wzou4rm3Tt#page=23).
>
> ## Q1. "This taxonomy might not capture more complex edits like scaffold hopping or ring system modifications"
> As suggested by the reviewer we conducted a scaffold changes analysis via Bemis–Murcko on 50,000 stratified random samples (95% confidence, ±1% error margin). The results show the coarse edit actions **indeed produce complex molecular transformations** (73-92% scaffold hops, 74% ring additions, ring deletions).
>
> | Action  | Valid Pairs | Same Scaffold | Scaffold Hop | Ring Add | Ring Del | Ring Mod |
> |---------|-------------|---------------|--------------|----------|----------|----------|
> | Delete  | 1,591       | 14.14%        | **85.86%**   | 0.13%    | **38.91%** | 60.97%   |
> | Insert  | 21,876      | 7.94%         | **92.06%**   | **74.22%** | 0.00%    | 25.78%   |
> | Replace | 26,533      | 26.96%        | **73.04%**   | 30.78%   | 3.49%    | 65.73%   |

---

> > ### Comment · Reviewer_wdBR · 2025-11-27
> >
> > Thank you for your response. My concern about the lack of evaluation across leading models and benchmarks is properly addressed. Despite the author's effort to test MEGA on three biological endpoints, this may not adequately demonstrate the usability of MEGA in real-world drug design due to the lack of test targets and comparison with domain-expert models. Collectively, I would like to change my score to a marginal negative (4).

---

### Official Review · Reviewer_13Xh · 2025-10-31

**Soundness:** 3
**Presentation:** 3
**Contribution:** 3
**Rating:** 6
**Confidence:** 2

**Summary:**

This paper introduces MEGA, a large-scale dataset for molecular editing and property-guided optimization. It contains over 31 million parent–child molecular pairs, each annotated with explicit edit actions (Insert, Replace, Delete) and property-based improvement labels. Using MEGA, the authors train a Llama-3 8B model via supervised fine-tuning (SFT) and Group Relative Policy Optimization (GRPO) with a similarity-aware reward. The MEGA-trained models significantly outperform prior datasets (MolEdit-Instruct, MolOpt-Instructions, DrugAssist) by up to +21.47pp in hit ratio and achieve a 36× gain in data efficiency under GRPO.

**Strengths:**

1. The paper is well-organized, clearly written, and easy to follow. Figures and tables support the narrative, and the methodology is presented with good clarity.
2. The experimental design is solid and thorough. The comparisons, ablations, and qualitative results convincingly demonstrate MEGA’s effectiveness and the robustness of the findings.
3. The MEGA dataset is large and diverse.

**Weaknesses:**

1. Because MEGA relies on predefined fragmentation rules (BRICS, HR, RECAP) and applies only one edit per molecule, it may overrepresent easy-to-fragment scaffolds and common functional groups while under-sampling complex chemistries such as macrocycles or multi-center transformations. The strong skew toward Replace and Insert actions (≈97%) could also bias models toward minimal, local edits and limit generalization to broader molecular modifications.

2. Property improvements are defined solely by RDKit-calculated proxies (LogP, QED, TPSA, etc.) and discrete thresholds. This could cause models to overfit to these proxy metrics rather than true pharmacological quality, and to exploit threshold boundaries. Similarly, the Tanimoto-based reward in RL favors small structural changes, which, while chemically valid, may discourage more creative or synthetically diverse optimizations.

**Questions:**

Please see weakness.

---

> ### Author Response · Authors · 2025-11-20
> **We thank Reviewer 13Xh for their thorough evaluation and address each concern below.**
>
> ## W1: "Reliance on fragmentation rules/local edits, thresholds and Tanimoto reward discourages Model’s creativity."
>
> We thank the reviewer for raising this important concern about model expressivity. We note that during Tanimoto-GRPO optimization (Section 4.2), the Tanimoto component of the reward is balanced with property hits. This allows for different levels of structural change, depending on the modifications needed to achieve the target property threshold.
>
> To assess whether this occurs in practice, we evaluated the MEGA-GRPO model using the ChatDrug benchmark (200 SMILES, 14 optimization tasks) for a total of **2,462 successful transformations**. The results (Table 1) show that MEGA-GRPO is able to perform chemically diverse modifications, including scaffold hops (40.5%), multi-site edits (22.7%), ring modifications (22.6%), and generation of macrocycles from non-macrocyclic inputs (32 examples).
>
> **Table 1: Chemical diversity analysis of MEGA-GRPO outputs on ChatDrug benchmark.**
>
> | Metric | Value |
> |--------|-------|
> | Total transformations | 2,462 |
> | Single-site edits | 77.3% |
> | Multi-site edits (≥2 sites) | 22.7% |
> | Bemis-Murcko Scaffold hops | 40.5% |
> | Ring modifications | 22.6% |
> | Macrocycles created (from non-macrocyclic inputs) | 32 |
>
>
> **Threshold boundaries analysis**: We computed the distribution of property improvements relative to task-specific thresholds. We observed that only 8.75% of successful cases cluster around thresholds (0-0.1σ), while 39% achieve large improvements exceeding 1σ above threshold. Regarding the failed cases, 61.7% moved in the correct direction but did not reach the thresholds. Furthermore, 87.5% of tasks follow non-normal distributions (Shapiro-Wilk, p < 0.05), indicating the model used different modification strategies during optimization.
>
>
> These patterns collectively suggest that the model produces meaningful and varied chemical transformations along property-relevant optimization trajectories, despite the Tanimoto-based similarity constraint. We added a dedicated section on the model's chemical edit behavior in [Appendix E.2](https://openreview.net/pdf?id=wzou4rm3Tt#page=28) of the updated manuscript.
>
>
> ## W2: "Models may overfit to RDKit proxies rather than true pharmacological quality."
>
> We thank the reviewer for raising this important point. The goal of a large-scale synthetic dataset such as MEGA is to teach LLMs robust and generalizable molecular optimization skills that can be transferred to other endpoints.
>
> To assess this in practice, we evaluated the zero-shot performance of MEGA-GRPO (Llama-3-8B post-trained with MEGA) on three biological endpoints that were never seen during training and cannot be computed with RDKit: Dopamine D2 Receptor (DRD2), c-Jun N-terminal Kinase 3 (JNK3), and Glycogen Synthase Kinase 3β (GSK3β). These endpoints require docking simulations or QSAR models, distinct from RDKit calculations.
>
> **Table 2: Zero-shot performance on biological endpoints.** SR = Success Rate, HR_L = Hit Rate (loose threshold), HR_S = Hit Rate (strict threshold).
>
> | Model             | Task   | SR%      | HR_L%    | HR_S%    |
> | ----------------- | ------ | -------- | -------- | -------- |
> | Llama-3-8B (MEGA) | DRD2↑  | **84.5** | **58.5** | **54.5** |
> | Llama-3-8B (BASE) |        | 9.5      | 4.5      | 1.0      |
> | Llama-3-8B (MEGA) | JNK3↑  | **81.0** | **29.5** | **27.5** |
> | Llama-3-8B (BASE) |        | 5.5      | 1.5      | 1.5      |
> | Llama-3-8B (MEGA) | GSK3B↑ | **83.5** | **51.0** | **48.5** |
> | Llama-3-8B (BASE) |        | 6.0      | 3.5      | 2.0      |
> | **Mean (MEGA)**   |        | **83.0** | **46.3** | **43.5** |
> | **Mean (BASE)**   |        | **7.0**  | **3.2**  | **1.5**  |
>
>
> As shown in Table 2, MEGA-GRPO exhibits strong zero-shot transfer: 83.0% mean success rate and 46.3% mean hit rate (loose threshold), representing +76.0pp and +43.1pp improvements over the base model, respectively.
>
> These results indicate that the model has not overfit to RDKit-specific endpoints or thresholds. Instead, it learned generalizable molecular optimization skills that transfer to previously unseen pharmacological endpoints. For further generalization evidence see [Appendix B](https://openreview.net/pdf?id=wzou4rm3Tt#page=22) in our updated manuscript.

---

### Official Review · Reviewer_BLFN · 2025-11-02

**Soundness:** 3
**Presentation:** 3
**Contribution:** 2
**Rating:** 4
**Confidence:** 4

**Summary:**

The paper outlines a new chemical edit dataset for use with large language models. The goal is to improve language models' understanding of consequences of chemical modifications such as replacing functional groups etc. Each dataset record includes a pair of smiles, edit operation used, as well as resulting property changes (whether they reach pre-defined thresholds). 28 tasks/properties included are all simple immediately calculable properties that RdKit can provide (e.g., aqueous solubility, QED, etc). The dataset is offered in two versions, small and large. The authors experiment with fine-tuning language models on their new dataset, including GRPO RL post-training, in comparison to other, smaller datasets, and showing clear gains in their dataset-aligned evaluation tasks. GRPO with a reward function adjusted with a thresholded Tanimoto similarity is shown to offer clear gains.

**Strengths:**

Distilling into language models some ability to understand consequences of chemical edits can be very helpful. The authors take a step in this direction by building a reasonably sized dataset (Mega-Large, 31M edit-pairs), derived from combinatorial compound library ZINC. The evaluation tasks are aligned with the dataset, including two-property combinations, showing (expectedly due to close alignment) gains in these tasks after fine-tuning with/without RL. The evaluation in this sense seems careful and comprehensive. The authors' choice of including some constructed "negative" pairs is a good addition to the dataset.

**Weaknesses:**

My main concern is that the properties in the dataset are RdKit calculable properties and thresholded with pre-set boundaries. These are relatively simple and straightforwardly calculable properties. If this is the focus, why not instead provide a virtual dataset (of any size) with calls to RdKit for properties rather than fix compounds, thresholds etc?

As far as I can see there are no measured quantities, no reaction information included in the proposed dataset. Reactions would seem like a more helpful fine-tuning dataset than edits. I.e., what products (including yield) would result from using a particular catalyst, reaction conditions, etc. Commercial databases such as Reaxys do include such information and thus may be more helpful for compound optimization, resolving retrosythetic pathways, or learning to hypothesize alternative, easily synthesizable products that have similar, desirable property profiles.

Given that edits, properties and thresholding are relatively simple, my concern is also that the language model improvements based on the dataset do not offer much practical utility, do not generalize beyond these edits. Some demonstration that the edit dataset helps generalize LLM capabilities beyond RdKit properties would be particularly helpful.

**Questions:**

Can you show any gains in tasks that are not directly aligned with tasks in the dataset?
Are allowable edit locations/types of edits also explicated in the dataset records?

---

> ### Author Response · Authors · 2025-11-17
> **We thank Reviewer BLFN for their thorough evaluation of MEGA and appreciate the constructive feedback. We address each concern below with additional clarification and experimental evidence.**
>
> ## W1: “Why not instead provide a virtual dataset for properties rather than fix compounds and thresholds”
> The value of MEGA is providing annotated pairs of local structure changes and their causal impact on target objectives. This cannot be achieved by computing RdKit descriptors alone. Our experimental results show that these demonstrations greatly increase the ability of LLMs to perform single-shot molecule optimization. MEGA required **~21 CPU-years** and **11.3B oracle calls** to exhaustively explore the ZINC250 edit space and generate 72M (31.4M positive, 41M negative) high-quality annotated examples. Re-generating anything comparable on the fly would be prohibitively expensive compared to simply using the provided dataset. Regarding the fixed tasks and thresholds, they ensure reproducibility and direct comparison with leading optimization methods (Liu et al., 2022; Wang et al., 2023; Ye et al., 2025). For further details on exploration efficiency see [Appendix A.2](https://openreview.net/pdf?id=wzou4rm3Tt#page=16) of our updated manuscript.
>
> ## W2: “Reactions would seem like a more helpful fine-tuning dataset than edits”
> Reaction information plays a crucial role in the synthesis and feasibility phase of the drug discovery pipeline. Our work focuses on the lead optimization stage. At this point, the core need is to train models to perform targeted, local edits toward specific objectives. A reaction dataset would not be of great benefit as it does not teach the model *how to locally modify a structure* towards a goal. That being said, MEGA can be combined with reaction datasets to teach more tasks to the model if desired.
> ## W3, Q1(a): “Model improvements won’t generalize beyond RdKit properties”
>
> We thank the reviewer for raising this relevant point and present evidence demonstrating that MEGA-trained models learn *core molecular editing skills* that transfer to properties and tasks not present in the dataset.
>
> **Biological properties:** We evaluate the zero-shot performance of MEGA-GRPO (Llama 3 8B post-trained with MEGA) on three biological endpoints that can not be computed with RDKit: Dopamine D2 Receptor (DRD2), c-Jun N-terminal Kinase 3 (JNK3), and Glycogen Synthase Kinase 3β (GSK3β). We follow the ChatDrug evaluation protocol (Liu et al., 2022) and report success rates and hit rates under strict and loose thresholds.
> | Model | Task | SR% | HR_L% | HR_S% |
> |-------|------|-----|-------|-------|
> | Llama-3-8B (MEGA) | DRD2↑ | **84.5** | **58.5** | **54.5** |
> | Llama-3-8B (BASE) |  | 9.5 | 4.5 | 1.0 |
> | Llama-3-8B (MEGA) | JNK3↑ | **81.0** | **29.5** | **27.5** |
> | Llama-3-8B (BASE) |  | 5.5 | 1.5 | 1.5 |
> | Llama-3-8B (MEGA) | GSK3B↑ | **83.5** | **51.0** | **48.5** |
> | Llama-3-8B (BASE) |  | 6.0 | 3.5 | 2.0 |
> | **Mean (MEGA)** | | **83.0** | **46.3** | **43.5** |
> | **Mean (BASE)** | | **7.0** | **3.2** | **1.5** |
>
> The substantial gains over the base model demonstrate that MEGA enables the learning of generalizable molecular editing patterns that extend beyond the RDKit endpoints used during training.
>
> **Held-out-tasks:** We further tested MEGA-GRPO on a set of multiple-objective optimization tasks entirely excluded from the training dataset. Consistent with the previous results, the post-trained version of Llama with MEGA greatly outperforms the base model in all tasks.
>
> | Model | Task | SR% | HR_L% | HR_S% |
> |-------|------|-----|-------|-------|
> | Llama-3-8B (MEGA) | LogP↑+TPSA↓ | **90.5** | **54.0** | **40.5** |
> | Llama-3-8B (BASE) |  | 2.0 | 0.0 | 0.0 |
> | Llama-3-8B (MEGA) | QED↑+TPSA↑ | **86.5** | **18.0** | **2.0** |
> | Llama-3-8B (BASE) |  | 6.0 | 0.0 | 0.0 |
> | Llama-3-8B (MEGA) | HBA↑+TPSA↑ | **94.0** | **92.0** | **82.0** |
> | Llama-3-8B (BASE) |  | 8.5 | 6.5 | 0.5 |
> | **Mean (MEGA)** | | **90.3** | **54.7** | **41.5** |
> | **Mean (BASE)** | | **5.5** | **2.2** | **0.2** |
>
> These findings collectively demonstrate that an LLM trained with MEGA exhibits generalizable molecule optimization capabilities beyond its training set. These results were included in [Appendix B](https://openreview.net/pdf?id=wzou4rm3Tt#page=22) of our updated manuscript.
>
> ## Q1(b): “Further details of the dataset records”
> Each MEGA record contains comprehensive edit information:
> a chemically valid edit operation (edit, insert, replace);
> the paired molecules before and after the edit;
> action-wise tracing of the modification (which fragment was replaced and with what, which fragment was removed, and the neighborhood in which an insertion took place). We added expanded dataset format details in [Appendix A](https://openreview.net/pdf?id=wzou4rm3Tt#page=14) of our updated manuscript.

---

### Meta-Review · Area_Chair_RpJ3 · 2026-01-03

**Summary:**

The paper proposes a large-scale molecular editing dataset for guided-action optimization, with total 72M molecule pairs. Experiments conducted on this dataset show great improvement.

The primary concerns shared by multiple reviewers focused on the dataset's reliance on computed RDKit proxies rather than experimental biological endpoints or reaction data, raising doubts about real-world utility in drug discovery. Reviewers questioned whether the model could generalize beyond these simple properties and if the predefined edit actions would limit chemical diversity and creativity. Additionally, there were initial requests for more rigorous benchmarking against state-of-the-art commercial and open-source models to prove the method's superiority.

**Reviewer Concerns:**

The authors addressed concerns regarding generalization and benchmarking by providing new zero-shot results on biological endpoints (e.g., DRD2, JNK3) and outperforming strong baselines like Gemini and DeepSeek on ChemCoTBench.
Concerns about limited chemical creativity were also largely resolved through additional analyses demonstrating significant scaffold hopping and macrocycle generation.

However, the fundamental concern regarding the dataset's synthetic nature (lacking experimental "wet lab" data) remains an outstanding philosophical disagreement for Reviewer wdBR, who believes this limits the practical applicability for real-world drug design.

**Reviewer Scores:**

Reviewer wdBR explicitly updated their score from 2 to 4, acknowledging the benchmark improvements but maintaining reservations about the lack of experimental targets, but still gives negative points.
Reviewer BLFN (initial 4) would likely increase their score, as their primary question regarding generalization beyond RDKit properties was answered with the biological endpoint experiments.

---

### Decision · Program_Chairs · 2026-01-26

Reject